# Implicit neural image field for biological microscopy image compression

Gaole Dai[1], Rongyu Zhang[1,4], Qingpo Wuwu[1,4], Cheng-Ching Tseng[1,4], Yu Zhou[2,4], Shaokang Wang[1,4], Siyuan Qian[1], Ming Lu[1], Ali Ata Tuz[3], Matthias Gunzer[2,3], Tiejun Huang [1] ✉, Jianxu Chen [2,5] ✉ & Shanghang Zhang [1,5] ✉

The rapid pace of innovation in biological microscopy has produced increasingly large images, putting pressure on data storage and impeding efficient data sharing, management and visualization. This trend necessitates new, efficient compression solutions, as traditional coder–decoder methods often struggle with the diversity of bioimages, leading to suboptimal results. Here we show an adaptive compression workflow based on implicit neural representation that addresses these challenges. Our approach enables application-specific compression, supports images of varying dimensionality and allows arbitrary pixel-wise decompression. On a wide range of real-world microscopy images, we demonstrate that our workflow achieves high, controllable compression ratios while preserving the critical details necessary for downstream scientific analysis.

Microscopic imaging is essential in modern biological research, not only for observational discovery but also for statistical analyses and mathematical modeling. The past three decades have witnessed great advances in imaging techniques and analysis methods in tandem. However, this progress has also presented a new challenge: the efficient storage and computational management of this massively increasing volume of microscopy data. While recent efforts in metadata standardization[1], data management tools[2] and next-generation file formats[3] have alleviated some challenges, the sheer size of microscopy data remains a concern.

To address this issue, we revisit a classic technique—image compression—which has not yet been systematically optimized for microscopy data. While commercial coder–decoder (CODEC) methods, such as JPEG2000[4] and high-efficiency video coding (HEVC)[5], offer some utility, they are not primarily designed for the unique characteristics of bioimages. For example, microscopy images could be of up to five dimensions, $X$ (length), $Y$ (width), $Z$ (depth), $C$ (channel) and $T$ (time), which could offer much more contextual information and therefore have much more potential redundancy to be compressed than standard RGB images or videos. Also, special optical properties for different light microscopes, such as the use of narrow-band filters in fluorescence

microscopy or the polarized light source in differential interference contrast microscopy, may cause the compression schemes designed for natural images to yield suboptimal results for bioimages.

Recent progress in artificial intelligence (AI) has yielded promising new compression techniques, one of which is based on implicit neural representation (INR)[6,7]. In brief, INR learns a sample-specific function using an artificial neural network (ANN) to map multidimensional spatiotemporal coordinates of any data to their corresponding pixel values. The core concept of INR focuses on representing data values through the parameter values of the neural network. On the one hand, works such as sinusoidal representation networks (SIREN)[7] systematically analyze how periodic activation functions enable neural networks to learn repetitive data structures more effectively. On the other hand, the flexibility of INR has inspired researchers to deploy this learning strategy across various applications, including three-dimensional (3D) reconstruction[8,9], registration[10] and super-resolution[11]. Recent advances have also uncovered the potential of INR-based compressors[12–17]. By limiting the parameter size of the ANN, supported with dimensional information, INR-based compressors theoretically enable controllable high compression ratios (CRs) and on-demand reconstruction of the full

[1]State Key Laboratory of Multimedia Information Processing, Peking University, Beijing, China. [2]Leibniz-Institut für Analytische Wissenschaften – ISAS, Dortmund, Germany. [3]Institute for Experimental Immunology and Imaging, University Hospital, University of Duisburg-Essen, Essen, Germany. [4]These authors contributed equally: Rongyu Zhang, Qingpo Wuwu, Cheng-Ching Tseng, Yu Zhou, Shaokang Wang. [5]These authors jointly supervised this work: Jianxu Chen, Shanghang Zhang. ✉e-mail: tjhuang@pku.edu.cn; jianxu.chen@isas.de; shanghang@pku.edu.cn

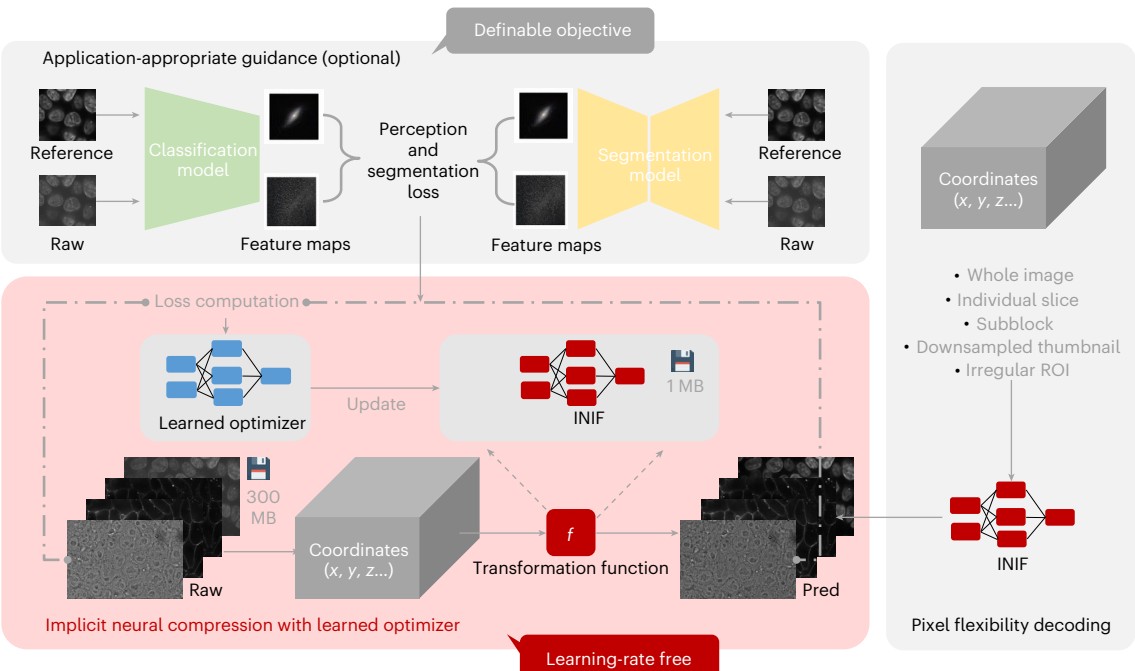

**Fig. 1 | The INIF compression–decompression workflow.** The general training process of INIF is shown in the red schematic, which begins by generating coordinates that have the same shape as the target compression images. These coordinates are then fed into the INR model to transform their corresponding pixel values. The target images are used as the ground truth for calculating the per-pixel similarity loss, which is optimized using a learned optimizer to update the INR model iteratively. Additional application-specific guidance, such as perceptual and segmentation loss, can be incorporated when necessary. The final storage content is a unique INIF file containing the weights of the INR model and the metadata. The INIF file can be decoded in flexible ways, including any individual slice, a collection of ROIs or a low-resolution version with GPU support. Pred, prediction.

data or any specific regions (even disconnected) of the data with a unified network. However, due to network size limitations, capturing intricate high-frequency details poses a great challenge for INR networks[18]. Recent INR-based compressors have attempted to borrow partition strategies from traditional CODECs, transforming data into smaller subblocks for individual compression. These methods[14–17] reported better compression quality. Still, the drawback of such subblock-specific learning is that it might require considerable time to compress a single image. At the same time, performance still largely depends on tuning certain training hyperparameters, such as the learning rate for each subblock. This issue is particularly pronounced for large bioimages.

In this work, we propose a new microscopy image compression paradigm, called implicit neural image field (INIF). INIF builds upon the effectiveness and flexibility of INR and addresses common bottlenecks in existing INR-based compression algorithms by adopting a learned optimizer[19]. Moreover, following the spirit of application-appropriate validation[20], INIF integrates application-specific guidance for improved compression quality and trustworthiness.

## Results

The overall workflow of our INIF microscopy image compression framework is illustrated in Fig. 1. The compression process is to train the INR network. The INR network is usually a small multilayer perceptron (MLP) with several linear layers. The inputs for this MLP are coordinate grids with identical shapes as the corresponding multidimensional microscopy data. The raw image serves as ground truth for calibrating the similarity between the reconstructed image and the raw image with a similarity loss, for example, mean square error, subsequently optimized by a learned optimizer[19], which does not need extensive searching for optimization hyperparameters, such as the learning rate (Figs. 2–4).

In addition to the core framework, we incorporated application-specific guidance by introducing supplementary loss functions beyond the default per-pixel similarity loss. In the 'Application-appropriate guidance with adjustable objective' section, we present two illustrative examples of such application-specific guidance: one aimed at preserving downstream segmentation performance (Fig. 5) and another focused on enhancing model robustness to noisy images obtained under conditions of very low laser power (Fig. 6). Furthermore, we demonstrate that utilizing CODEC results as prior guidance can lead to more efficient compression. The INR network functions as an adapter to reconstruct the residual images, which represent the discrepancy between the original image and the CODEC-compressed version (Extended Data Fig. 1).

The compression performance was compared against a representative commercial CODEC, HEVC and a representative INR-based baseline method, SIREN[7], with visual qualitative demonstration and additional similarity metrics. It is critical to know whether the compression quality is sufficient, which depends on the downstream tasks, and users could confirm the validity via application-appropriate validations[20] before deployment.

After the compression process, the model weights and shape information are stored on disk. To decompress, a grid of the full image size or a collection of certain grid points can be fed into the INR network to retrieve the full image or any subparts of the full image. Unlike classic CODECs, such as JPEG, PNG or TIFF, where the full image is always decoded as a whole, or more sophisticated file formats, such as Zarr-based formats[21], which allow chunk-wise access or retrieval of precalculated low-resolution thumbnails, INIF offers ultimate flexibility by enabling efficient retrieval of any region of interest (ROI) or even subsampled previews. For example, it can retrieve one pixel every four grid points along the X dimension for 4× downsampling, with similar options for the Y and Z dimensions (Extended Data Fig. 2). Such flexibility in decompression holds the potential to be integrated into large-scale analysis, browser-based big image visualization.

## Compression of microscopy data of different dimensionalities

**Volumetric microscopy images (*XYZ*).** Volumetric light microscopy images are widely used in biology, considering the 3D nature of biological samples. Common two-dimensional image compression CODECs

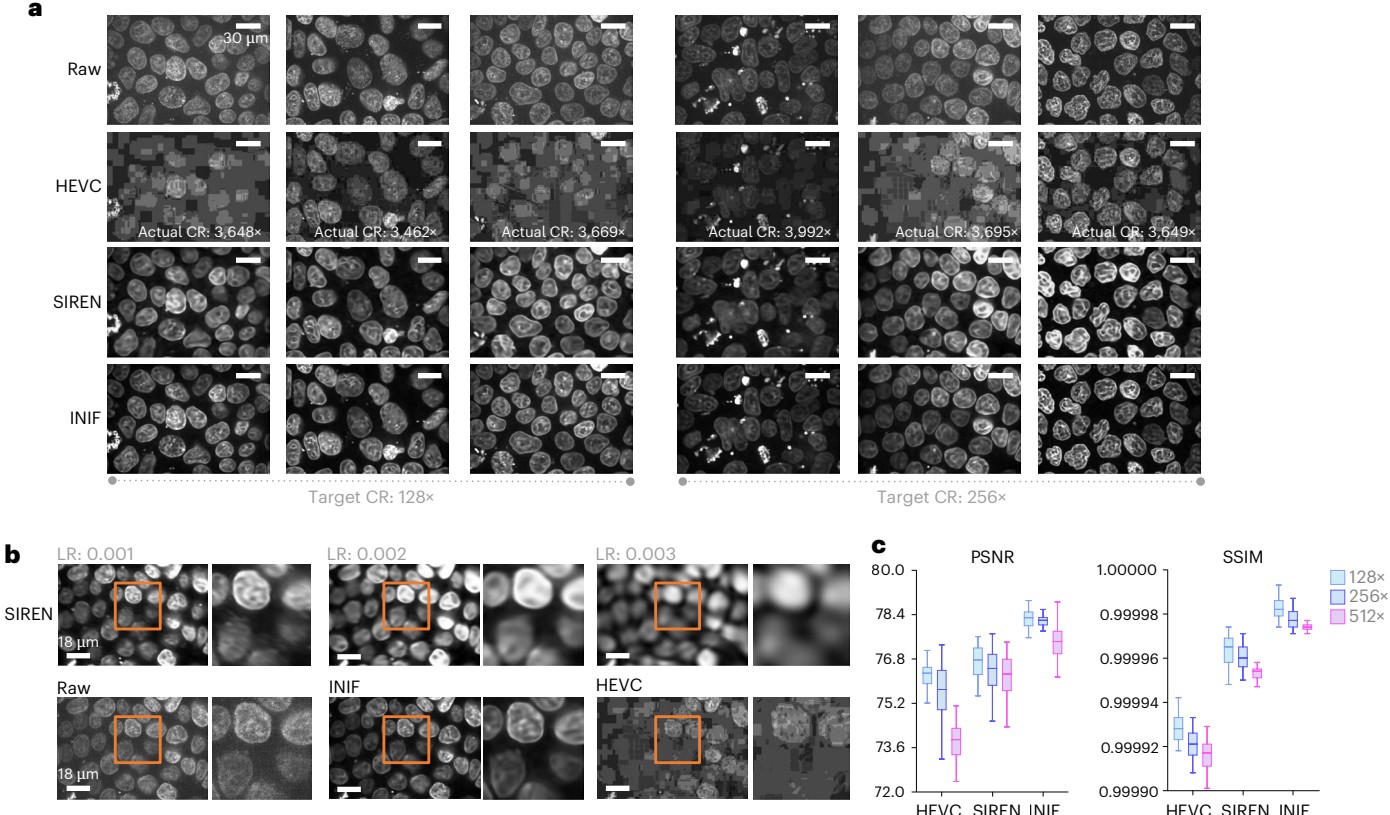

**Fig. 2 | Comparison of INIF, HEVC and SIREN on 3D hiPS cell data. a**, The visualization results demonstrate that HEVC exhibits uncontrollable actual CR and noticeable blocking artifacts (row 2). SIREN, as a pure INR method, successfully satisfied the target CR and captures the overall contour of the nucleus but lacks a detailed reconstruction of its edges and inner contents (row 3). By contrast, INIF effectively maintains clear visibility of cellular contours and intranuclear material distribution and retains the CR controllability even under high CRs, 128× and 256× (row 4). **b**, SIREN is sensitive to the setting of the learning rate (LR) (row 1), while INIF generates sharper compressed images without LR optimization (row 2, middle). **c**, Quantitative assessment of restoration error using PSNR (left) and SSIM (right) metrics (higher is better) across different CRs (128×, 256× and 512×) for $n = 25$ samples. Data are presented using a standard box plot where the center line represents the mean, the box indicates the first and third quartiles and the whiskers extend to the largest and smallest data values within 1.5 times the interquartile range. We crop out and zoom in the content in orange box to highlight the detail quality after using different methods for compression.

without taking the lateral context into account could be problematic, while the time dimension for common video compression CODECs has a different assumption than the lateral dimension of 3D microscopy due to special optical properties. We demonstrate the effectiveness of INIF compression on a 100× high-resolution 3D confocal spinning disk microscopy image of human induced pluripotent stem (hiPS) cells[22]. Example results are visualized in Fig. 2. We can observe that images compressed with HEVC exhibit substantial blocking artifacts, which hinder the discernment of nucleoli and chromatin within cell nuclei. Furthermore, even with meticulous adjustment of hyperparameters such as bitrate[23], constant rate factor[24] and adaptive quantization mode[25], it is unable to control the CR throughout the HEVC experiment. By contrast, our proposed INIF method consistently maintains clear visibility of cellular contours and retains considerable textural details under controllable high CRs to the target 128- and 256-fold. Meanwhile, unlike SIREN—which is sensitive to training hyperparameters and requires tuning with various learning rates to improve quality, resulting in additional computational cost when compressing large datasets—INIF achieves higher-quality compression without the need for learning rate tuning. We further assessed the restoration error by comparing decompressed images with ground-truth images across three distinct CRs (128×, 256× and 512×) in Fig. 2c. We observed that INIF outperformed HEVC and SIREN in both the peak signal-to-noise ratio (PSNR) and structural similarity index (SSIM)[26] by a substantial margin.

**Multiplexed microscopy images (*XYC*).** Multiplex microscopy, such as multiplexed immunofluorescence[27], enables simultaneous detection and colocalization analysis of multiple markers, making it an indispensable tool for uncovering the heterogeneity in biological studies. Here, the additional information along the channel dimension makes the common CODECs suboptimal. We verified the performance of INIF on multiplexed immunofluorescence imaging of breast tumor tissues from patients with triple-negative breast cancer (TNBC)[28]. Each sample includes five channels: S100A4-Opal 520, CK-Opal 570, PDPN-Opal 650, 4′,6-diamidino-2-phenylindole (DAPI) and bright field with 3,506 × 3,506 pixels along *XY* (see 'Multichannel breast tumor data' section in Methods). Specifically, S100A4 and PDPN are proteins that exhibit high expression in cancer-associated fibroblasts, while Cytokeratin (CK) stains the epithelial regions and DAPI stains adenine–thymine-rich regions in DNA. By merging the S100A4 or PDPN channel with the CK or DAPI channel, colocalization analysis within the corresponding microenvironment can be performed[28]. The objective, in this case, is to preserve cancer-associated fibroblast heterogeneity after compression. The results are shown in Fig. 3. Upon visualization, it is evident that HEVC produced unsatisfactory results, where the presence of blocking artifacts substantially affects the final quality. As observed in the volumetric microscopy image example, SIREN shows a similar issue—its performance is sensitive to the learning rate, and changes to the learning rate can lead to corrupted results.

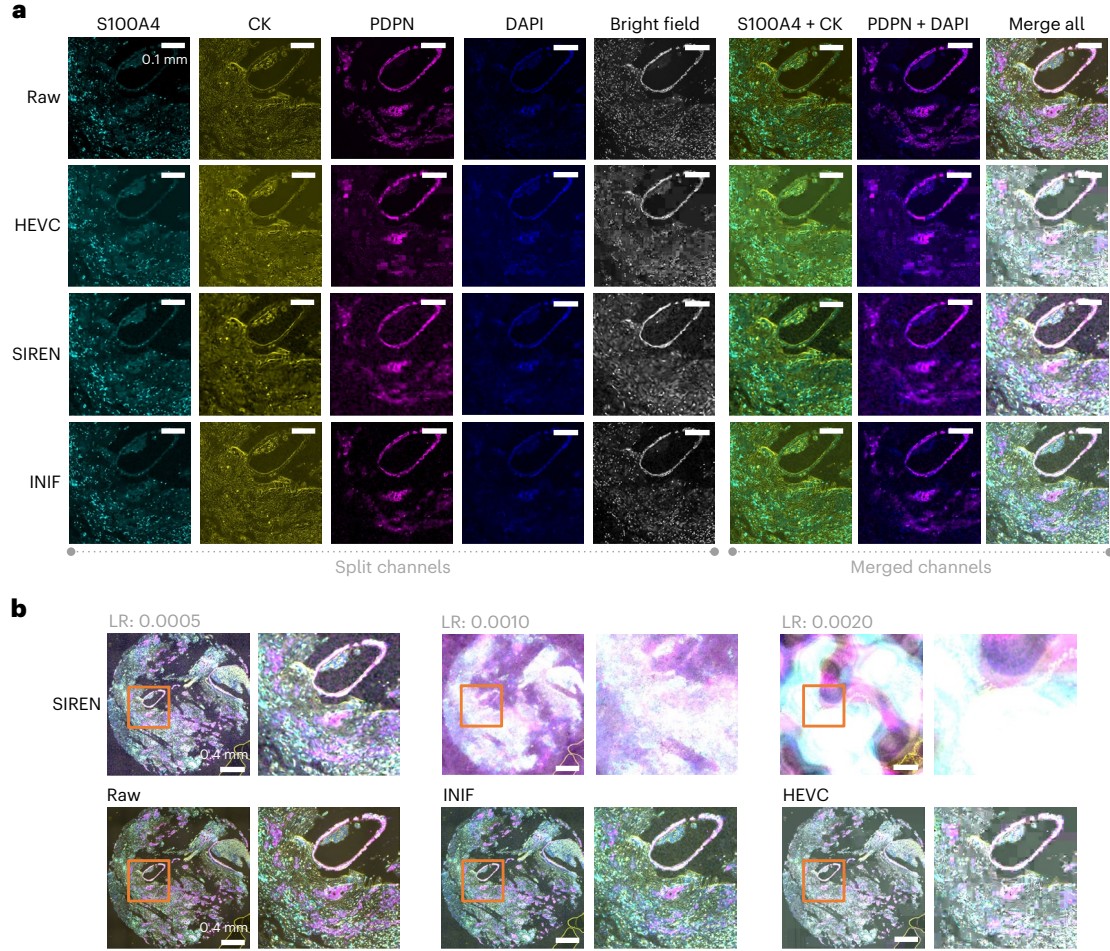

**Fig. 3 | Comparison of INIF, HEVC and SIREN on multichannel breast tumor data. a**, The target CR is 512× for all methods. Although HEVC achieved CR control in this experiment (511×), it still yields unsatisfactory results with noticeable blocking artifacts, especially visible in the merged visualization (row 2). SIREN exhibits issues of over-smoothing, compromising the quality of merged outcomes, with a relatively poor brightness relationship maintained between the channels (row 3). Meanwhile, INIF demonstrates more satisfactory results in both individual and merged outputs than the others (row 4). **b**, SIREN exhibits sensitivity to the LR, leading to undesired results with minor changes (top row). We crop out and zoom in the content in orange box to highlight the detail quality after using different methods for compression.

**Multichannel volumetric time-lapse microscopy images (*XYZTC*).** Commercial CODECs can not be directly applied to the full image with more than three dimensions. We could use such a CODEC to independently compress the data block by block, which completely loses the contextual information. Consequently, such fragmentation leads to a noticeable deterioration in the quality of the resulting compressed images. Conversely, the biggest challenge faced by partition-free methods, such as SIREN, is the issue of underfitting, which arises when a limited number of parameters is used to model a vast data space. This motivated us to enhance the expressiveness of the network using better optimization strategies, adopting learned optimizers to overcome the gradient stagnation. We demonstrate the applicability of INIF in five-dimensional (5D) microscopy images, with the volumetric mouse organoids imaged with dual oblique plane microscopy[29] over time. The 5D movies showcased the growth of mouse organoids. The primary objective of this dataset is to discern the 3D dynamics in the membrane and DNA channels over time during mitosis. It is crucial to maintain the cellular shape and count even after compression. Upon analyzing the results obtained under a target CR of 256 (Fig. 4), we have observed that controlling the final CR of HEVC is challenging due to the need to align numerous individual compression processes. The boundaries between cells in decompressed images using HEVC have become indistinguishable, thereby obscuring the precise localization

and 3D structure of cells within the organoid. The results from SIREN exhibit visible alterations in granular structures in the membrane and DNA. By contrast, INIF not only maintains a stable CR but also preserves an exceptional level of detail. This preservation enables clear differentiation of individual cell outlines using just one single INR network.

**Application-appropriate guidance with adjustable objective**
The compression process in INIF can be extended with additional guidance according to specific downstream tasks or requirements. The guidance is achieved via optimization with task-specific loss. In this section, we demonstrate two different application-appropriate guidance examples: one example where we aim to optimize the compression to maintain the accuracy in a downstream segmentation task, and another example where we aim to reduce the noisy signals in the process of compression.

Automatic image segmentation is a common downstream task in quantitative biology studies. We hypothesize that making the compression process aware of the specific downstream segmentation task could help preserve critical—and potentially very subtle—information necessary for accurate segmentation, even if that information is not essential for visualization. For demonstration, we tested the compression on 3D confocal microscopy (100×) of Troponin I Type 1 (TNNI1) in cardiomyocytes[30] (see 'TNNI1-stained hiPS cell 3D data' section

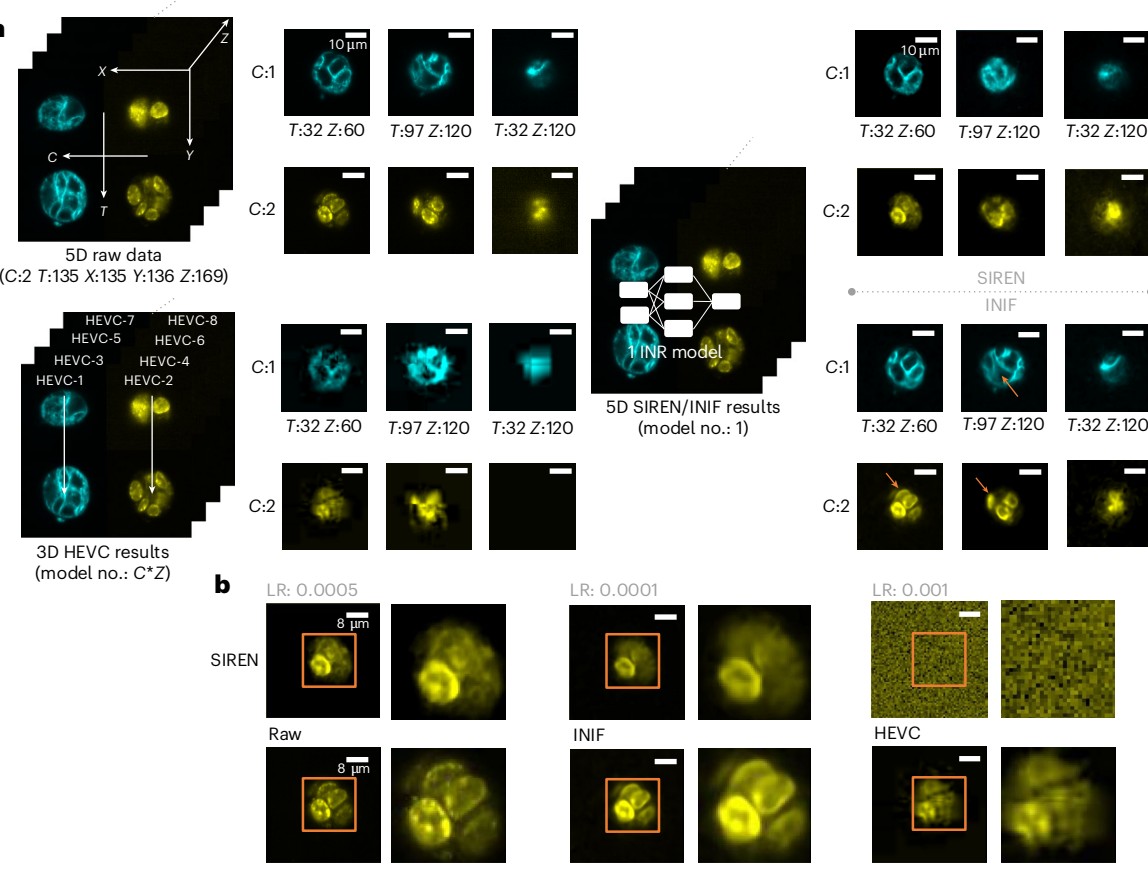

**Fig. 4 | Comparison of INIF, HEVC and SIREN on 5D mouse organoid data.**
**a**, The target CR is 256× for all methods. HEVC struggles to control the final CR (64×) and produces decompressed images with merged cell boundaries, obscuring the precise localization and 3D structure of cells (bottom left). Although both SIREN (top right) and INIF (bottom right) maintain a consistent CR (256×), only INIF preserves exceptional detail (errors in SIREN are indicated

by orange arrows in INIF). This enables clear differentiation of individual cell outlines and facilitates vivid observation of dynamic changes in cells over time. **b**, The SIREN model still relies on selecting an appropriate LR range for training, as the performance is greatly affected by distributions that fall outside of this range. We crop out and zoom in the content in orange box to highlight the detail quality after using different methods for compression.

in Methods). TNNI1 displays a distinct filamentous morphology with striped patterns. As shown in Fig. 5, across various decompressed outcomes using INIF, HEVC and SIREN techniques, the overall filamentous shape can be preserved, but with very different performance in preserving the striped patterns. For example, SIREN tends to produce excessively smoothed images that lack these finer details, while the HEVC approach introduces a substantial amount of noise into the output. By contrast, INIF effectively retains the striped patterns to permit further sarcomere analysis[31]. This is further supported by segmentation results on the decompressed images, where INIF achieved balanced performance with minimal false positives and false negatives.

From the results above, the INR-based compression method can mitigate high-frequency noise within the data to a certain extent, compared with classic CODECs, such as HEVC. This phenomenon can be attributed to the inherent characteristic of neural networks, which tend to learn continuous functions[32], particularly in compression tasks where model size is constrained. In this context, noise represents a challenging target for modeling purposes[33]. Therefore, we demonstrate the flexibility of INIF to simultaneously compress the data and reduce the noise, that is, only the important signals are compressed. These noisy biological microscopy images are fairly common when imaging live cells with low laser power. To this end, INIF can be coupled with an additional perceptual loss function. Specifically, during compression, we incorporated additional reference data from relatively clear samples into each training iteration by selecting two random areas: one from the reconstruction by the INR network and another from the clear reference data. These two areas were then passed through the same

classification network[34] to obtain distinct perception feature maps. Subsequently, we calculated the perceptual loss between two feature maps as part of a joint optimization process aiming to simultaneously improve pixel-level similarities to noisy raw data while considering perception-level similarities to clear reference data. To verify the performance, we tested on confocal microscopy recordings of developing *Tribolium castaneum* (red flour beetle) embryos with different laser powers: low and very low[35]. As shown in Fig. 6, the results indicate, that when faced with noticeable noise at low laser power, HEVC exhibited severe block artifacts, particularly on structures with lower contrast. Without perceptual guidance, INIF could suffer from similar information loss as in SIREN. The perceptual loss encourages INIF to focus on the most critical information and, therefore, shows better generalization capability to noisy data.

## Faster INIF with CODEC priors

Despite the limitations of CODEC in achieving optimal compression results in bioimaging, its advantage of minimal compression costs, in terms of both time and computation[5], compels us to explore the possibility of incorporating it into the INIF framework to improve compression efficiency. The primary advantage of the aforementioned framework is the versatility and effectiveness in compression. However, iterative training unavoidably slows down the compression process. To address this issue, INIF has a hybrid mode that incorporates classic CODEC when fast compression is important. Specifically, in this mode, the initial compression attempt is performed using any CODEC, such as HEVC. Typically, this approach is cost-effective as these CODECs require

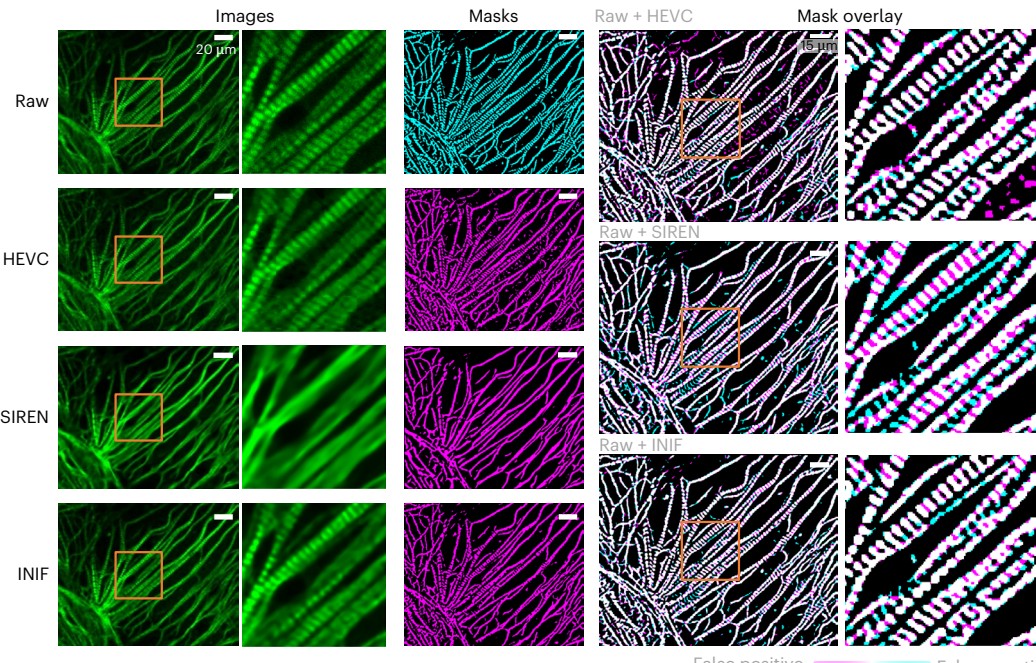

**Fig. 5 | INIF with segmentation guidance on TNNI1 data.** The target CR is 256× for all methods. The optimal outcome should demonstrate the precise overlap between the magenta (mask after compression) and cyan masks (reference mask), resulting in a binary mask of black–white (true negative and true positive). Suboptimal outcomes would exhibit two colors: cyan values indicate missed segmentation by excluding existing areas with decompressed data (false negative), while magenta values represent incorrect segmentation by including nonexisting areas with decompressed data (false positive). The segmentation results show that INIF achieves balanced outcomes compared with HEVC and SIREN. HEVC tends to produce false-positive results, while SIREN suffers from oversmoothed artifacts, leading to false negatives. We crop out and zoom in the content in orange box to highlight the detail quality after using different methods for compression.

only a few iterations to complete the compression process without training. Then, we transform our INIF backbone into an adapter[36] by substituting the raw image ground truth with residuals obtained by subtracting CODEC-decompressed images from raw images. By conducting a short additional training phase using INIF, we can compare the adapted result with the original result to determine if overwriting the current slice is necessary. It should be noted that, in this case, we still recommend an application-appropriate selection standard; in our experiment, we selected centrosome count as an application-specific metric. For more general usage scenarios, selection based on common metrics such as PSNR or SSIM would also be feasible. With this design, we not only enable adjustable compression objectivity to reduce blocky artifacts associated with CODEC but also expedite the training process of INIF by reducing information load.

We demonstrated the advantages of the hybrid mode of INIF by testing on 3D 100× confocal microscopy images of the Centrin-2 (CETN2) protein marking the centrosome in hiPS cells[22]. The most distinctive morphological feature of CETN2 in the image is isolated or pairs of bright blobs that are specifically localized to the centrioles varying at different cell cycle stages. Our analysis revealed that HEVC compression obscured critical puncta. Within a comparable amount of compression time, SIREN-compressed images suffered from incomplete convergence of the loss function due to limited training epochs, leading to substantial blurring that hindered practical application. However, INIF leveraged CODEC priors to expedite compression and achieved clearer images even with a short training period (Extended Data Fig. 1). See Supplementary Sections 3.4–3.7 for more comprehensive benchmarking results.

## Discussion

The primary objective of INIF is to raise awareness of efficient bioimaging data storage and to provide a practical solution within existing hardware infrastructure. With INIF, large-scale microscopy data can be compressed at a high and controllable CR while being stored in a single INIF file that supports flexible, pixel-wise decoding. Validated through extensive experiments, INIF advances the state-of-the-art image compression by achieving higher CRs, superior reconstruction quality and adaptable objectives throughout the differentiable compression pipeline.

However, INIF is still a lossy compression method and cannot preserve all information without any information loss. For example, when compressing breast tumor tissue data (Fig. 3), although we achieved superior visualization results compared with HEVC and SIREN, it remained challenging to retain all informative signals, especially under higher CRs. Therefore, we strongly recommend the use of lossless compression techniques[37] if compressed data cannot meet the required standards for downstream analysis. In addition, compared with traditional CODECs, INR-based methods involve iterative optimization through gradient descent using optimizers, which leads to slower compression speeds on the same hardware. In the future, additional experiments benchmarking the efficiency of INIF on different hardware, such as different graphics processing units (GPUs), cloud servers or even edge devices, could further elucidate the potential of integrating INIF in different deployment scenarios (for example, on edge devices attached to imaging instruments or a dedicated computing cluster of a core imaging facility).

## Methods

### Architectures of the INR network and learned optimizer

The SIREN network is a simple MLP with a cosine activation function. For all our experiments, we utilized a seven-layer SIREN[7] as the backbone of our INR network. The exact model size is decided according to the target CR and the shape information. The core of this training process starts with generating coordinate grids with identical shapes as the corresponding multidimensional microscopy data. These coordinates

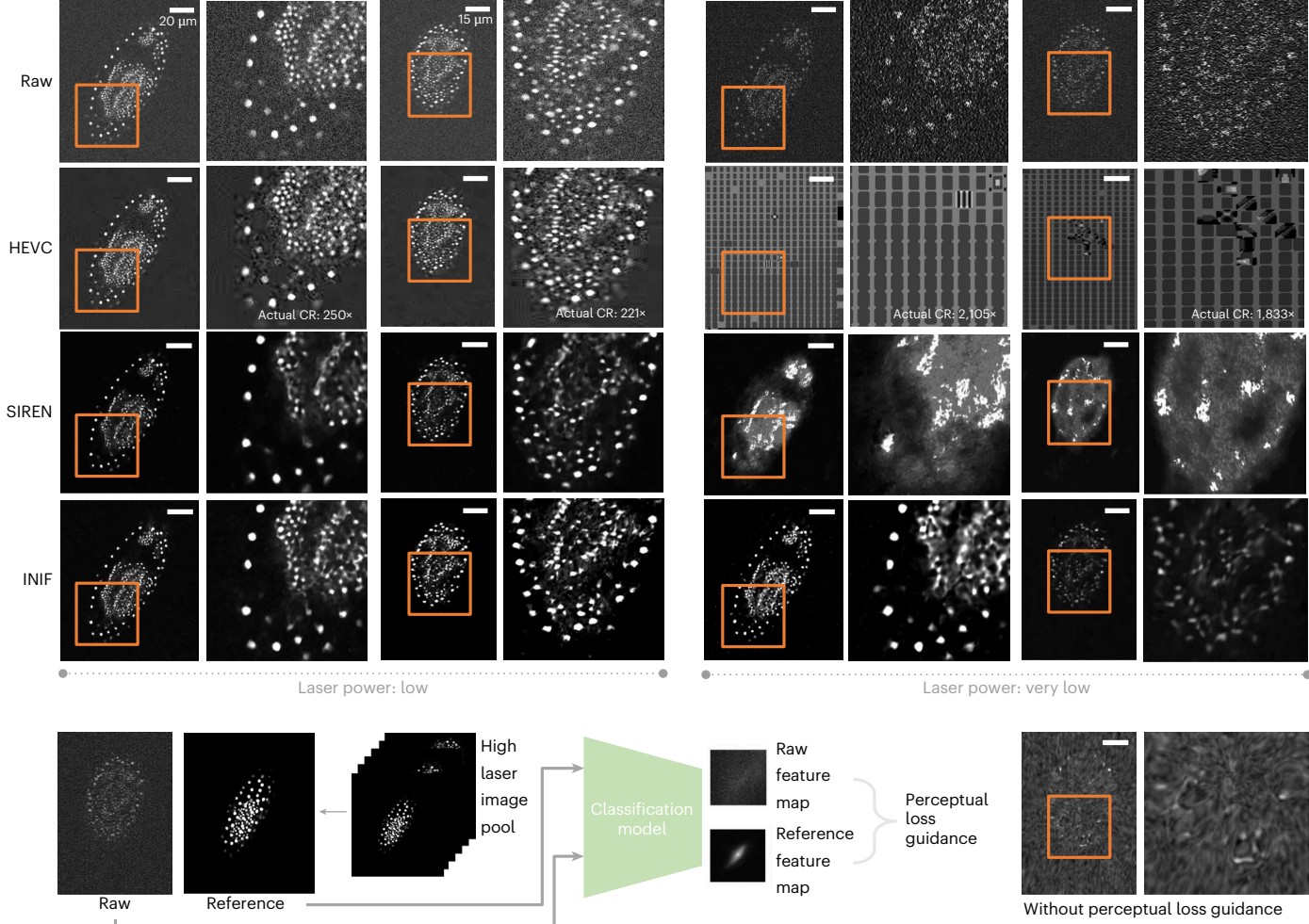

**Fig. 6 | INIF with perceptual guidance on noisy red flour beetle embryo data.** The target CR is 256× for all methods. HEVC exhibits severe block artifacts, particularly on structures with lower contrast (left row 2), while SIREN and INIF perform well in both high-contrast and low-contrast structures (left rows 3 and 4). When dealing with extremely noisy data captured using very low laser power settings, HEVC confronted an uncontrollable CR and failed to reconstruct meaningful results. SIREN and INIF preserved the overall structure under target CR; however, INIF with perceptual-level guidance achieves better robustness compared with HEVC, SIREN and INIF without perceptual guidance. We crop out and zoom in the content in orange box to highlight the detail quality after using different methods for compression.

are utilized as inputs and fed into the INR network to transform their respective pixel values. We made some minor adjustments, such as introducing a learnable cosine frequency and refining the initialization strategy. One of our key innovations lies in the adoption of learned optimizers, which leverage ANNs to update ANNs instead of relying solely on hand-crafted algorithms. In this study, we used the versatile learned optimizer VeLO[19], which has been pretrained at scale for optimization tasks, and we further fine-tuned it specifically for compression tasks. Our algorithm was implemented using the Google JAX framework[38]. See Supplementary Section 2 for more details.

## Data prepossessing
The initial step involves normalizing the input images to ensure consistency in their range. We have implemented min–max normalization[39], which entails defining the minimum and maximum intensity values and subsequently mapping all values linearly within this range. Specifically, we have set the intensity range from 0 to 100. Before training, another preprocessing step is performed wherein we derive coordinates by extracting shape metadata from the target data and initializing coordinates with identical shapes corresponding to the data. Furthermore, these coordinates are normalized within each dimension to a range of −1 to 1. See Supplementary Section 2.4 for more details.

## Segmentation loss
We utilize the Allen Cell Structure Segmenter (Segmenter)[30] to extract the segmentation map from images. The workflow of Segmenter involves several steps for image preprocessing and segmentation, including intensity normalization, edge-preserving smoothing, 3D filament filtering and removal of small objects. Parameters for contrast enhancement were set as [1.5, 10.5], edge-preserving smoothing intensity was set to 1, F3 Params for filament detection were set as [1, 0.01] and remove small object intensity was set to 20. The segmentation loss is calculated using the intersection over union (IoU) loss between the segmentation results of original and decompressed images. IoU can be formulated as follows:

$$\text{IoU} = \frac{\text{Intersection}}{\text{Union}} = \frac{M_{\text{original}} \cap M_{\text{decompressed}}}{M_{\text{original}} \cup M_{\text{decompressed}}}. \tag{1}$$

## Perceptual loss
The classification model chosen for extracting perceptual feature maps is the learned perceptual image patch similarity (LPIPS)[34], which quantifies the discrepancy between the reference and reconstructed data. Specifically, we utilize the AlexNet version of LPIPS, which has been pretrained on a natural image classification task. To further enhance

its performance on bioimaging data, we fine-tune the model using a small amount of slice data from the hiPS cell dataset.

During the noisy data compression task, we randomly select a batch of pixel values from the clear reference and reconstruct an equal number of pixel values using the INR network. These two random image patches are then passed through the fine-tuned LPIPS model. The perceptual loss is defined as follows:

$$\mathcal{L}_{p} = \sum_{i=1}^{L} \frac{1}{H_i W_i} \sum_{H_i, W_i} \| w_i (\Phi_i(\hat{y}) - \Phi_i(y)) \|_2^2. \tag{2}$$

In this context, $\mathcal{L}_{p}$ represents the perceptual loss, $\hat{y}$ refers to the reconstructed patch, $y$ represents the reference patch, $\Phi_i(\cdot)$ denotes the feature representation at layer $i$, and $w_i$ is used as a scale factor to normalize each feature map in fine-tuned LPIPS model.

## Statistics and reproducibility

For each of the described experiments and case studies, we (1) randomly extracted suitable data from private or public datasets (no statistical method was used to predetermine sample selection), (2) trained an INR network using a learned optimizer with or without additional guidance based on the desired functionality to be demonstrated and (3) used the trained INR network for decoding to decompress the data, subsequently quantifying and reporting the obtained results. The experiment was repeated three times independently.

## Compression with CODEC prior

The HEVC standard[5] is specifically designed to provide an efficient encoding solution for transmitting high-definition video content. However, the absence of differentiable operators in HEVC impedes its adaptability to biological microscopy. Here, we provide an extra methodology by using INIF as an adapter[36]. Specifically, we allocate 90% of the bitstream for preliminary compression using HEVC, followed by computing the residual image by differencing between the original image and the decompressed output from HEVC. The remaining 10% of the bitstream is utilized to initialize INIF. See Extended Data Fig. 1 and Supplementary Sections 2.3 and 3.5 for more details.

## Decoding strategies of INIF

Commonly used file formats (for example, JPG and PNG) can only be used for regular two-dimensional RGB images. Specialized file formats such as Ome-TIFF or Zarr have been developed for storing and decoding high-dimensional microscopy images[21,40,41]. Compared with Ome-Tiff, Zarr[21] offers more flexibility with additional features such as multiresolution storage, block-wise decoding and parallel decoding using a GPU. The evolution of these file formats indicates that the ultimate objective is to efficiently decode and visualize specific ROIs[42] in any microscopy data as needed, because full decoding can be time-consuming and memory-intensive. In this context, INIF provides an excellent solution by enabling pixel-wise decoding. Taking a 3D (*XYZ*) confocal microscopy image of DNA[22] as an example, assume our goal is to visualize slice 32. Decoding TIFF requires complete decoding of the entire dataset with dimensions *X*:924, *Y*:624 and *Z*:65, including extracting all pixel values, followed by selecting only those belonging to slice 32 for visualization purposes. Meanwhile, leveraging INIF's pixel-wise decoding function allows us to simply provide the coordinates of slice 32 to the INR network for efficient decoding. Furthermore, by assigning a subset of coordinates, we can also support multiresolution functionality, particularly useful when fast preview-based decodings are required without needing additional storage space for different resolution versions of the same data, which is used in Zarr format. In addition, even in special scenarios involving irregularly shaped ROIs represented by binary masks, we can still handle them through appropriate location-based decoding (Extended Data Fig. 2).

## 3D DNA-stained hiPS cell data

hiPS cells are a type of stem cell artificially generated in the laboratory by reprogramming adult somatic cells to return to an embryonic stem cell-like state. The WTC-11 hiPSC Single-Cell Image Dataset v1 encompasses 25 distinct intracellular structures observed in hiPS cells, with each sample including a channel specifically stained for DNA[22]. We randomly selected 25 samples exhibiting diverse structures and compressed the DNA-stained channel. See Supplementary Table 2 for more details.

## Multichannel breast tumor data

We utilized a 3D (*X, Y, C*) multichannel fluorescence microscopy dataset of breast tumor samples obtained from the 1-TNBC dataset within the Image Data Resource public dataset project ID 2801[28]. The 1-TNBC dataset comprises 200 breast tumor tissue samples. Each sample represents a multidimensional image with the following characteristics: (1) spatial dimensions of 4,506 × 4,506 pixels depicting the two-dimensional tissue structure, (2) five channels labeling distinct cellular structures and biomarkers in the tumor samples, namely S100A4-Opal 520 image (labeling S100A4 protein), CK-Opal 570 image (labeling cytokeratins), PDPN-Opal 650 image (labeling podoplanin), DAPI image (labeling cell nuclei) and bright-field image. In our experiment, we utilized the Patient_55_core_1.tiff file in Fig. 3.

## 5D mouse organoid data

This is a 5D (*X, Y, Z, C, T*) multichannel fluorescence microscopy dataset that captures the growth dynamics of mouse organoids over time. It was obtained from the Zenodo repository at ref. 29. The dataset consists of six tiles (Tile_1 through Tile_6) containing time-lapse four-dimensional image sequences. Each tile represents a multidimensional image with specific properties: (1) spatial dimensions of 135 × 136 × 160 representing the volumetric structure in three dimensions; (2) two fluorescence channels labeling DNA (red channel) and cell membrane (green channel); and (3) 135 timepoints capturing the temporal evolution of organoid morphogenesis. For visualization and validation purposes, we utilized a subset of the Tile_1_processed_binned-2b file by extracting 12 subvolumes. See Supplementary Table 3 for more details.

## TNNI1-stained hiPS cell 3D data

These data are four-dimensional (*X, Y, Z, C*) multichannel fluorescence microscopy data that comprise the 3D structural data of the TNNI1 protein obtained from fluorescently tagged hiPS cell lines at the Allen Institute for Cell Science[30]. This dataset represents a multidimensional image with specific properties: (1) spatial dimensions of 555 × 441 × 60 representing the volumetric structure in three dimensions; (2) four fluorescence channels: DNA channel, cell membrane channel, TNNI1 protein structure channel and bright-field channel. For visualization and validation purposes, we utilized the TNNI1 protein structure at depth 30.

## Noisy *Tribolium* data

This dataset comprises confocal microscopy recordings of developing *Tribolium castaneum* (red flour beetle) embryos, obtained from ref. 35, with three levels of laser power: high, low and very low. All images were acquired using the Zeiss 710 multiphoton laser scanning microscope equipped with a 25× multiimmersion objective. In our experiment, we evaluate the performance of our proposed method on two samples captured at low and very low laser power from this dataset. See Supplementary Table 4 for more details.

## Centrin-2-stained hiPS cell 3D data

In this experiment, we utilized the images (the structure channel) of the CETN2 cell line from the WTC-11 hiPSC Single-Cell Image Dataset v1[22] to investigate its association with the centrosome. The centrosome, composed of two centrioles surrounded by pericentriolar material, plays a crucial role in cell division by duplicating and aiding in the formation of the mitotic spindle for proper chromosome segregation.

**Reporting summary**

Further information on research design is available in the Nature Portfolio Reporting Summary linked to this article.

## Data availability

The microscopy images used in the article were all obtained from the public dataset, downloaded according to the instructions provided in each corresponding reference. Brief descriptions of these public datasets, including the reference, are listed in Methods. From all these public data, we uploaded the collection of exact image data files used for the experiments via Zenodo at https://doi.org/10.5281/zenodo.16412927 (ref. 43) for reproducibility purposes. See Supplementary Section 1 for more details. Source data are provided with this paper.

## Code availability

Code to reproduce the experiments in this work or to compress your own data is available via GitHub at https://github.com/PKU-HMI/INIF and via Zenodo at https://doi.org/10.5281/zenodo.16746747 (ref. 44). The license use in the repository follows Apache License Version 2.0.

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

## Acknowledgements

G.D. was supported by the National Natural Science Foundation of China (grant no. W2442028). S.Z. was supported by the National Science and Technology Major Project (grant no. 2022ZD0117800). The work of ISAS was supported by the 'Ministerium für Kultur und Wissenschaft des Landes Nordrhein-Westfalen' and 'Der Regierende Bürgermeister von Berlin, Senatskanzlei Wissenschaft und Forschung'. Y.Z. and J.C. were further supported by the Bundesministerium für Forschung, Technologie und Raumfahrt (BMFTR) under the funding reference 161L0272. The work of M.G. was also partially supported by the Deutsche Forschungsgemeinschaft (DFG) through research grants GU768/10-1, GU769/15-1, GU769/15-2 and TRR332 C6.

## Author contributions

G.D. designed the main methodology, performed the primary experiments and wrote the original draft. R.Z., C.-C.T., Q.W., Y.Z. and S.W. contributed to parts of the investigation, algorithm development and writing. S.Q., M.L., A.A.T. and M.G. assisted with data collection and curation and provided guidance on the biomedical application and paper editing. T.H., J.C. and S.Z. supervised the research, acquired funding, administered the project and reviewed and edited the final paper. All authors reviewed and approved the final paper.

## Funding

V.

## Competing interests

The authors declare no competing interests.

## Additional information

**Extended data** is available for this paper at https://doi.org/10.1038/s43588-025-00889-4.

**Correspondence and requests for materials** should be addressed to Tiejun Huang, Jianxu Chen or Shanghang Zhang.

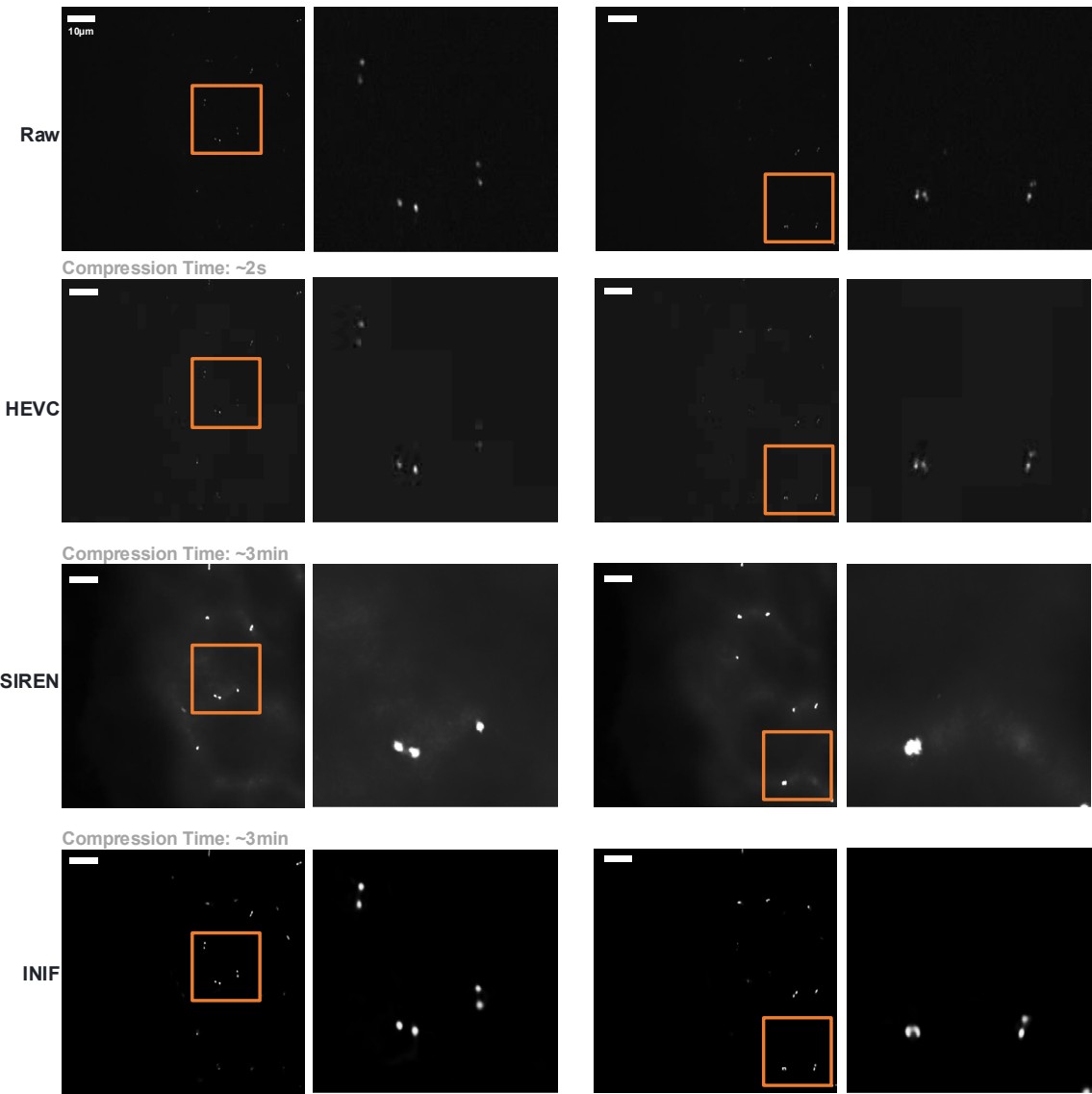

**Extended Data Fig. 1 | INIF with CODEC prior.** The target compression ratio is 256x for all methods. HEVC compression results obscured critical dot-like details (row 2). SIREN results suffer from incomplete convergence, leading to blurring (row 3). INIF leverages CODEC prior and achieves clearer images with a short training period (row 4).

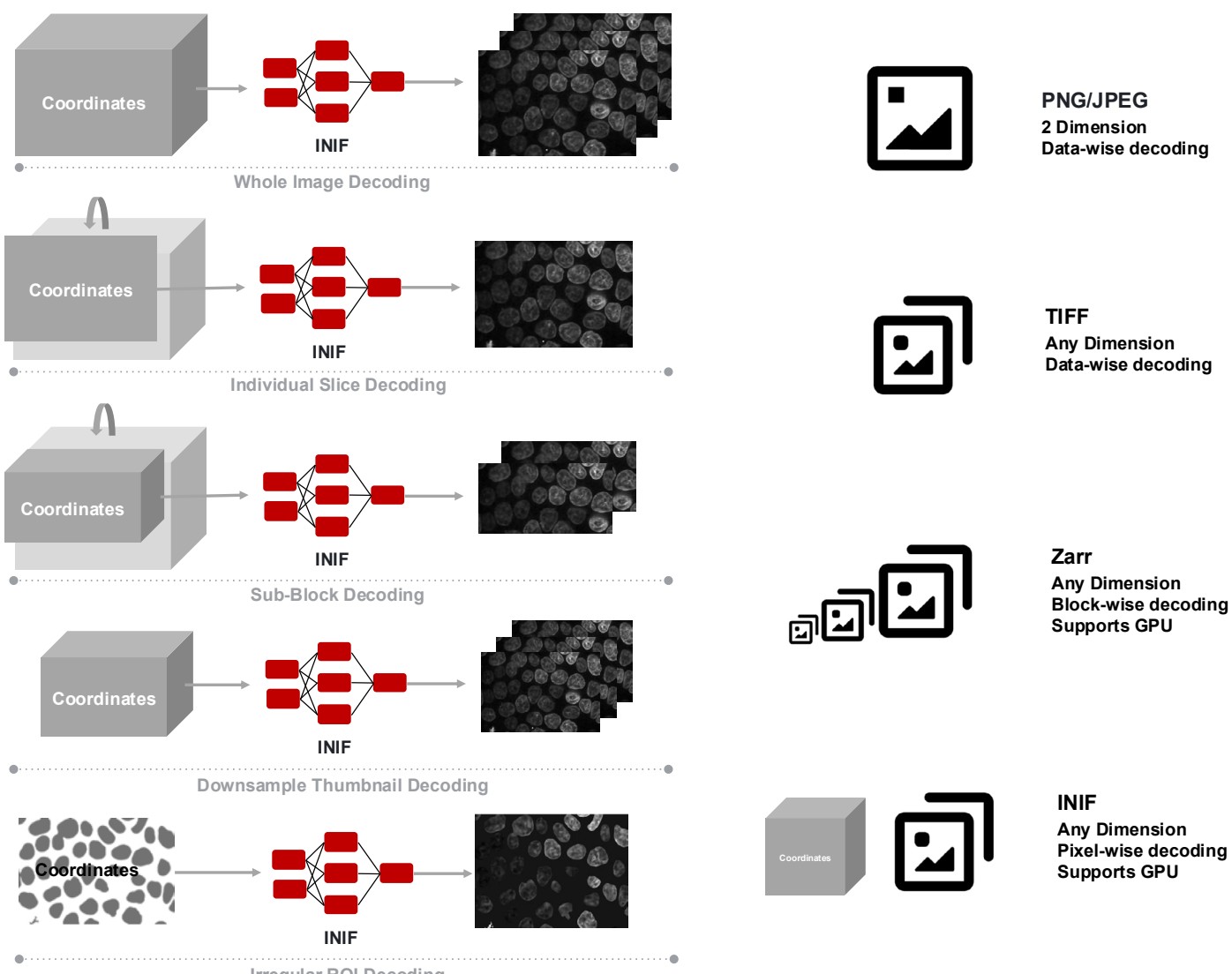

**Extended Data Fig. 2 | Decoding INIF file in a pixel-wise fashion.** The target compression ratio is 256x for all methods. Unlike traditional file formats like TIFF that require complete decoding of the entire dataset, INIF enables pixel-wise decoding by providing the coordinates of the desired region of interest (ROI) to the INR network. INIF also supports multi-resolution decoding functionality and location-specific decoding even for irregularly shaped ROIs represented by binary masks.

# Reporting Summary

## Statistics

For all statistical analyses, confirm that the following items are present in the figure legend, table legend, main text, or Methods section.

| n/a | Confirmed | |
|---|---|---|
| ☐ | ☒ | The exact sample size (*n*) for each experimental group/condition, given as a discrete number and unit of measurement |
| ☐ | ☒ | A statement on whether measurements were taken from distinct samples or whether the same sample was measured repeatedly |
| ☒ | ☐ | The statistical test(s) used AND whether they are one- or two-sided<br>*Only common tests should be described solely by name; describe more complex techniques in the Methods section.* |
| ☒ | ☐ | A description of all covariates tested |
| ☒ | ☐ | A description of any assumptions or corrections, such as tests of normality and adjustment for multiple comparisons |
| ☐ | ☒ | A full description of the statistical parameters including central tendency (e.g. means) or other basic estimates (e.g. regression coefficient) AND variation (e.g. standard deviation) or associated estimates of uncertainty (e.g. confidence intervals) |
| ☒ | ☐ | For null hypothesis testing, the test statistic (e.g. *F*, *t*, *r*) with confidence intervals, effect sizes, degrees of freedom and *P* value noted<br>*Give P values as exact values whenever suitable.* |
| ☒ | ☐ | For Bayesian analysis, information on the choice of priors and Markov chain Monte Carlo settings |
| ☒ | ☐ | For hierarchical and complex designs, identification of the appropriate level for tests and full reporting of outcomes |
| ☒ | ☐ | Estimates of effect sizes (e.g. Cohen's *d*, Pearson's *r*), indicating how they were calculated |

*Our web collection on statistics for biologists contains articles on many of the points above.*

## Software and code

Policy information about availability of computer code

| Data collection | All data used in the main text are from public data repositories, e.g., from previous publications. |
|---|---|
| Data analysis | all data being analyzed in the manuscript were performed using the provided code (shared via Github, https://github.com/PKU-HMI/INIF) |

For manuscripts utilizing custom algorithms or software that are central to the research but not yet described in published literature, software must be made available to editors and reviewers. We strongly encourage code deposition in a community repository (e.g. GitHub). See the Nature Portfolio guidelines for submitting code & software for further information.

## Data

Policy information about availability of data

All manuscripts must include a data availability statement. This statement should provide the following information, where applicable:
- Accession codes, unique identifiers, or web links for publicly available datasets
- A description of any restrictions on data availability
- For clinical datasets or third party data, please ensure that the statement adheres to our policy

All data used in the main text are from public data repositories, e.g., from previous publications. We have included the data availability section in the manuscript to provide all details.

## Human research participants

Policy information about studies involving human research participants and Sex and Gender in Research.

| | |
|---|---|
| Reporting on sex and gender | n/a |
| Population characteristics | n/a |
| Recruitment | n/a |
| Ethics oversight | n/a |

Note that full information on the approval of the study protocol must also be provided in the manuscript.

# Field-specific reporting

Please select the one below that is the best fit for your research. If you are not sure, read the appropriate sections before making your selection.

☒ Life sciences  ☐ Behavioural & social sciences  ☐ Ecological, evolutionary & environmental sciences

For a reference copy of the document with all sections, see nature.com/documents/nr-reporting-summary-flat.pdf

# Life sciences study design

All studies must disclose on these points even when the disclosure is negative.

| | |
|---|---|
| Sample size | All training were done per sample due to the specific algorithm developed in the paper, i.e., the same displayed in the manuscript are the entire sample collection we used, as our compression method will train one model per sample. |
| Data exclusions | no |
| Replication | we provide full details in the repository to reproduce our results shown in the paper. |
| Randomization | Samples were not randomized, as single samples were used to generate algorithm training data. The selection of samples from the public data repository was random. |
| Blinding | Investigators were not blinded to sample specifications, as the experiments do not include sample group comparisons. |

# Reporting for specific materials, systems and methods

We require information from authors about some types of materials, experimental systems and methods used in many studies. Here, indicate whether each material, system or method listed is relevant to your study. If you are not sure if a list item applies to your research, read the appropriate section before selecting a response.

## Materials & experimental systems

| n/a | Involved in the study |
|---|---|
| ☐ | ☒ Antibodies |
| ☒ | ☐ Eukaryotic cell lines |
| ☒ | ☐ Palaeontology and archaeology |
| ☐ | ☒ Animals and other organisms |
| ☒ | ☐ Clinical data |
| ☒ | ☐ Dual use research of concern |

## Methods

| n/a | Involved in the study |
|---|---|
| ☒ | ☐ ChIP-seq |
| ☒ | ☐ Flow cytometry |
| ☒ | ☐ MRI-based neuroimaging |

## Antibodies

| | |
|---|---|
| Antibodies used | This information is only related to the stress test in the supplementary information (Appendix C.1)<br>Light sheet fluorescence microscopy:<br>CD31, MEC13.3 clone, Biolegend, Cat. No:102528, 1:100<br>EpCAM (CD326), G8.8 clone, Biolegend, Cat. No:118212, 1:100 |

CD3, 17A2 clone, Biolegend, Cat. No:100209, 1:100
CD19, REAfinity, Miltenyi Biotec, Cat. No:130-131-141, 1:50

Validation

This information is only related to the stress test in the supplementary information (Appendix C.1)
All antibodies were bought from commercial suppliers as stated in the reporting summary. Widely used antibody clones and companies were selected, single- and multi-color stainings were made to verify filter settings during imaging. Validations supplied by the manufacturers are also available in the respective websites reached by the catalog numbers.

# Animals and other research organisms

Policy information about studies involving animals; ARRIVE guidelines recommended for reporting animal research, and Sex and Gender in Research

Laboratory animals

C57/BL6/J 8-12 weeks old wildtype mice for LSFM imaging experiments. Mice were housed in individually ventilated cages with a 12-hour dark/night cycle. Housing rooms have 21-23°C temperature and 40-60% humidity.

Wild animals

The study did not use wild animals.

Reporting on sex

Only female animals were used for LSFM imaging studies.

Field-collected samples

The study did not use field-collected samples.

Ethics oversight

Ethical approvals were obtained from the local authority, Landesamt für Natur, Umwelt und Verbraucherschutz Nordrhein-Westfalen, under permission number G1719/19 and conducted in accordance with the ARRIVE guidelines.

Note that full information on the approval of the study protocol must also be provided in the manuscript.

