## [Peer Review File · Nature Computational Science]

Implicit Neural Image Field for Biological Microscopy Image Compression

Corresponding Author: Dr Jianxu Chen

This manuscript has been previously reviewed at another journal that is not operating a transparent peer review scheme. The manuscript was considered suitable for publication without further review at Nature Computational Science.

A version of this paper was originally rejected for publication by Nature Computational Science, however that decision was reconsidered after appeal by the authors.

Version 0:

Decision Letter:

** Please ensure you delete the link to your author homepage in this e-mail if you wish to forward it to your co-authors. **

Dear Dr Chen,

Your manuscript "Implicit Neural Image Field for Biological Microscopy Image Compression" has now been seen by 3 referees, whose comments are appended below. You will see that while they find your work of interest, they have raised points that need to be addressed before we can make a decision on publication.

The referees' reports seem to be quite clear. Naturally, we will need you to address *all* of the points raised.

While we ask you to address all of the points raised, the following points need to be substantially worked on:

- The core idea of compressing images of any dimension and enabling arbitrary pixel-wise decompression using implicit neural representation is well-documented in the literature. Therefore, please provide a qualitative and quantitative comparison with these methods.
- The literature review should be more comprehensive.
- Comparing compressed images with significantly different ratios is not ideal and should be rectified.
- Please discuss the difference between INIF and INR.
- Please discuss how it is ensured that essential information in the residual would not be lost after compression by network again.
- Please discuss if the use neural network to learn pixel values corresponding to coordinates for different region sizes robust for transformations.
- Please discuss if the method also sensitive to learning rates, similar to the compared SIREN algorithm.
- Manuscript lacks the proof that this method working for microscope images. Please provide a complete data comparison of compression times for residuals/raw-image and SIREN.
- The reported tests do not include speed comparisons with other techniques, and additional details on this aspect are necessary.
- The results should also be evaluated in terms of overall energy consumption and the environmental impact of the proposed approach, especially when image compression is translated into a resource-intensive computational process.
- The comparison between different lossy compression methods is primarily qualitative. Metrics such as PSNR and SSIM are not provided for all tests, and additional details should be included.

Please use the following link to submit your revised manuscript and a point-by-point response to the referees' comments (which should be in a separate document to any cover letter):

Link Redacted

** This url links to your confidential homepage and associated information about manuscripts you may have submitted or be reviewing for us. If you wish to forward this e-mail to co-authors, please delete this link to your homepage first. **

To aid in the review process, we would appreciate it if you could also provide a copy of your manuscript files that indicates your revisions by making use of Track Changes or similar mark-up tools. Please also ensure that all correspondence is marked with your Nature Computational Science reference number in the subject line.

In addition, please make sure to upload a Word Document or LaTeX version of your text, to assist us in the editorial stage.

To improve transparency in authorship, we request that all authors identified as 'corresponding author' on published papers create and link their Open Researcher and Contributor Identifier (ORCID) with their account on the Manuscript Tracking System (MTS), prior to acceptance. ORCID helps the scientific community achieve unambiguous attribution of all scholarly contributions. You can create and link your ORCID from the home page of the MTS by clicking on 'Modify my Springer Nature account'. For more information please visit www.springernature.com/orcid.

We hope to receive your revised paper within three weeks. If you cannot send it within this time, please let us know.

Best regards,

Ananya Rastogi, PhD
Senior Editor
Nature Computational Science

Reviewers comments:

Reviewer #1 (Remarks to the Author):

Dai et al. present an implicit neural image field method for compressing biological microscopy images. The method combines a learned small neural network (MLP) with a learned optimizer to improve upon current learning-based image compression baselines, enabling fast adaptation to different images.

While the approach shows promise, the technical contributions could benefit from further development. The core idea of compressing images of any dimension and enabling arbitrary pixel-wise decompression using implicit neural representation or coordinate-based neural representation (CNR) from NeRF [1], as well as the faster adaptation of training the INIF model using a learned optimizer, are already well-documented in the literature. This overlap makes it challenging for the manuscript to stand out. The optional components in Fig. 1 are impressive, yet they are not the central contributions of this work. Thus, in its current form, the manuscript lacks sufficient novelty and contribution to the community.

Additionally, the literature review could be more comprehensive. The manuscript does a good job of introducing INR methods for natural images as inspiration for their application to biological images. However, it would be beneficial to include a more thorough discussion of more recent works. Since the publication of Sitzmann et al. in 2020, many studies have aimed to enhance compression performance from various perspectives. There is a wealth of related work in fields such as NeRF and CNR image compression (e.g., [2-4]). The authors are strongly encouraged to include more literature on CNR image compression, particularly those that improve training speed and decoding performance ([2-6]). Including a discussion that distinguishes this work from closely related studies would also be beneficial.

The experimental comparison section could be more complete and fair. This completeness is closely tied to the literature review; a thorough review would lead to more profound experimental comparisons. Another concern is the fairness of comparisons to HEVC, especially in Fig. 2. Although HEVC can be challenging to control for fixed compression ratios, comparing compressed images with significantly different ratios is not ideal. This discrepancy makes it difficult for readers to fully appreciate the impact of the proposed method.

Some technical details need clarification. For instance, how is the compression ratio calculated? Does it include the network configuration, network weights, and latent code? Are we training an INIF for each individual image stack? What are the preprocessing steps for the raw images to be compressed?

Minor Points:

- Please line-number the manuscript for easier referencing.

- [1] Mildenhall, Ben, Pratul P. Srinivasan, Matthew Tancik, Jonathan T. Barron, Ravi Ramamoorthi, and Ren Ng. "Nerf: Representing scenes as neural radiance fields for view synthesis." *Communications of the ACM* 65, no. 1 (2021): 99-106.
- [2] Tancik, Matthew, Ben Mildenhall, Terrance Wang, Divi Schmidt, Pratul P. Srinivasan, Jonathan T. Barron, and Ren Ng. "Learned initializations for optimizing coordinate-based neural representations." In *Proceedings of the IEEE/CVF Conference on Computer Vision and Pattern Recognition*, pp. 2846-2855. 2021.
- [3] Ladune, Théo, Pierrick Philippe, Félix Henry, Gordon Clare, and Thomas Leguay. "Cool-chic: Coordinate-based low complexity hierarchical image codec." In *Proceedings of the IEEE/CVF International Conference on Computer Vision*, pp. 13515-13522. 2023.

[4] Kim, Hyunjik, Matthias Bauer, Lucas Theis, Jonathan Richard Schwarz, and Emilien Dupont. "C3: High-performance and low-complexity neural compression from a single image or video." In Proceedings of the IEEE/CVF Conference on Computer Vision and Pattern Recognition, pp. 9347-9358. 2024.

[5] Blard, Théophile, Théo Ladune, Pierrick Philippe, Gordon Clare, Xiaoran Jiang, and Olivier Déforges. "Overfitted image coding at reduced complexity." arXiv preprint arXiv:2403.11651 (2024).

[6] Sheng, Xihua, Chuanbo Tang, Li Li, Dong Liu, and Feng Wu. "NVC-1B: A Large Neural Video Coding Model." arXiv preprint arXiv:2407.19402 (2024).

Reviewer #2 (Remarks to the Author):

****Summary****

The authors propose a novel method that combines image compression with meta-learning optimizer for compressing microscopy data. This approach preserves important information in image and allows for further validation in downstream tasks. The method not only utilizes the residual between the output of existing CODEC algorithms and the original data as training data to accelerate the process, but also incorporates the characteristics of downstream segmentation tasks to guide the training loss. Additionally, the use of meta-learning to optimize the optimizer avoids issues related to hyper-parameter selection. The method also offers high flexibility by allowing the selection of different regions of interest (ROIs, or patches) for compression rather than compressing the entire image firstly.

****Pros****

1. The combination of image compression with meta-learning and its application to microscopy data is a novel idea.
2. The ability to focus on any sub-region rather than a fixed full image provides high flexibility and strong practicality.

****Comments****

1. What is the difference between INIF and INR? Is it just the addition of a meta-learning optimizer?
2. "This residual compression mechanism can significantly reduce the compression time since most of the information is compressed by the fast codec and the amount of information that the INR network needs to learn is greatly reduced." If most of the information is compressed by the fast codec and only the residual, which contains information we need (details), is left, does further compressing the residual help retain more details? How can we ensure that essential information in the residual would not be lost after compression by network again?
3. The idea of using a neural network to learn pixel values corresponding to coordinates for different region sizes is innovative. However, is the network robust for transformations? For example, if the original data undergoes affine transformations, such as translation or rotation, would different coordinates affect the results significantly? So do we need to strictly align all the data to some standard?
4. Is this method also sensitive to learning rates, similar to the compared SIREN algorithm, which is why meta-learning was used? Have other meta-learning algorithms been tested for comparison?
5. Most importantly, it lacks the proof that this method working for microscope images. Only a very small proportion in a microscope image is decision-correlated, yet the method does not aim at those particular areas. The paper only provides visualizations for a subset of examples. Are there any comparisons of compression quality across different datasets in terms of metrics like PSNR and MS-SSIM or other metrics? Additionally, is there a complete data comparison of compression times for residuals/raw-image and SIREN? The article mentions using downstream segmentation tasks in training loss for guidance, but lacks sufficient demonstration of how well the compression retains important information. May need to show how high is the quality of the retained information based on downstream task validation.

Reviewer #3 (Remarks to the Author):

In their work, the authors introduce an efficient compression approach based on implicit neural representation, producing promising quality results. While the research is both original and engaging, I do not find it sufficiently innovative to warrant publication in Nature Computational Science for the following three key reasons.

First, as the authors themselves acknowledge throughout the paper, the method requires a considerable amount of time to compress a single image. The reported tests do not include speed comparisons with other techniques, and additional details on this aspect are necessary. This raises significant concerns regarding the scalability of the method for processing very large datasets.

Second, the training of the INR network demands substantial computational resources. This calls into question the benefit of improved data storage efficiency if it requires such significant computational investment. In my view, the results should also be evaluated in terms of overall energy consumption and the environmental impact of the proposed approach, especially

when image compression is translated into a resource-intensive computational process.

Third, the comparison between different lossy compression methods is primarily qualitative. Metrics such as PSNR and SSIM are not provided for all tests, and additional details should be included. Furthermore, to ensure a comprehensive quantitative comparison, other metrics reflecting the impact on segmentation or classification parameters should be considered.

Finally, I offer a few minor suggestions:

- I recommend that the authors provide complementary work on the point spread function of the imaging system in Figure 7.
- The size of the images in Figures 2, 3, 4, and 6 should be increased, as the differences between them are often difficult to discern.

Version 1:

Decision Letter:

Dear Dr Chen,

Your manuscript "Implicit Neural Image Field for Biological Microscopy Image Compression" has now been seen by 3 referees, whose comments appear below. In the light of their advice, we have decided that we cannot offer to publish your manuscript in Nature Computational Science.

From the reports, you will see that while they find your work of some potential interest, the referees raise concerns about the advance your findings represent over earlier work and the strength of the novel conclusions that can be drawn at this stage. In particular, they raise concerns about missing comparisons with SOTA, use of limited microscopy images, and the nuclei touching problem.

Indeed, it seems that a significant amount of additional work and clarification would be necessary in order to address the aforementioned points, and it is not clear whether or not your present results will remain strong in light of this additional work. Therefore, we feel that these criticisms are sufficiently important as to preclude publication of your work in Nature Computational Science.

Should future experimental data allow you to address all points put forth by reviewers, we would be willing to look at a revised manuscript and potentially reconsider the work at our discretion (unless, of course, something similar has by then been accepted at Nature Computational Science or appeared elsewhere). This includes submission or publication of a portion of this work somewhere else.

If you are interested in submitting a suitably revised manuscript in the future or if you have any questions, please contact me. In case you are not interested in submitting a revised version, you may transfer your manuscript to another journal in the Nature Portfolio using the link I provide at the end of this email.

I am sorry that we cannot be more positive on this occasion, but hope that you find the referees' comments helpful when preparing your paper for resubmission elsewhere.

Best regards,

Ananya Rastogi, PhD
Senior Editor
Nature Computational Science

Reviewers' Comments:

Reviewer #1 (Remarks to the Author):

While providing clarifications and benchmarking the compression model's performance, the revised manuscript still lacks the following essential clarification:

1. The poor performance of HEVC when targeted to a 256x compression ratio is caused by non-optimal codec settings. It is unfair to compare directly with HEVC results compressed by thousands of folds. It is not scientific to compare apples to oranges due to the massively different compression ratios presented by different methods.

2. HEVC is not the most advanced codec for biomedical image compression. With film grain analysis, Duan et al. (<https://www.biorxiv.org/content/10.1101/2024.07.17.603794v1>) showed that AV1 codec can compress similar quality nucleus images by up to 12,000x with visually much more faithful results than the 256x compressed results presented in this manuscript.

3. The statement that using a model that allows pixel-level image synthesis will lead to faster I/O performance when handling large-scale images is misleading. This is because when the image is large, the model size will be exponentially increased to compensate for the image dimensions. As it always requires the whole model to reconstruct even a single pixel, the time and memory required for loading the whole model can be slow or even impossible.

Reviewer #2 (Remarks to the Author):

Thanks for addressing some of previous comments. However, there are still many major problems remaining.

1. The images presented from Figure 2 to Figure 7 are all the demonstrations of particular cases. The profile and visualization of the whole dataset are missing. Moreover, quantified results compared with SOTA methods are missing. For example, in Section 1.2.3, the manuscript says 'using simple metrics such as PSNR or SSIM often contains bias and can be misleading when dealing with biomedical data', but there is only one segmentation task evaluated on Dice. Many more experiments should be performed and presented with charts or figures.
2. The innovation in this paper is the contour-aware compression algorithm, as contour recognition is essential in microscopy images. However, it cannot cope with the problems of touching or overlapping nuclei issues or nuclei clusters, that are usually found in microscopy images. The nuclei touching problem is tough to address in biomedical microscopy image analysis, yet this compression algorithm does not notice it. Moreover, INIF incorporates application-appropriate guidance. The authors should demonstrate the effectiveness of guidance with the ablation study.
3. The compression performance is subject to the capability of nuclei segmentation in the proposed method. Moreover, inner areas are also of high importance.
4. The author uses a minimal amount of microscopy images and of very limited types, i.e., "a public dataset of mouse organoids imaged with dual oblique plane microscopy", "100x high-resolution 3D confocal spinning disk microscopy image of human induced pluripotent stem cells (hiPSC)" and "a public dataset comprising MxIF imaging of breast tumour tissues from triple-negative breast cancer (TNBC) patients". This is more applicable to journals like IEEE Transactions of Medical Imaging. A general model should take in many more datasets. For example, "nnU-Net" published in 2021 Nature Method used "23 public datasets used in international biomedical segmentation competitions." The biomedical dataset boosted in the past 3 years.
5. The compression ratio (128X, 256X) is compared with the original raw images. The compression rate is not compared with any SOTA methods. Theoretically, the compression rate is already very high with any prevalent compression method, particularly for microscope images with large black background. For example, a thorough comparison with established end-to-end autoencoder compression algorithms, such as BMS, CST, EVC, and MLIC+, is needed. Comprehensive rate-distortion performance metrics and comparisons are required to validate the effectiveness. Specifically, the authors presented 'This dataset serves as the benchmark test for both SCI [11] and TINC [12] and they are representative state-of-the-art INR-based compressors for specialized biomedical data'. Yet the comparison is missing in Table 4. The performance gains from the INIF adapter appear heavily contingent on the use of specific CODECs like HEVC. Please compare the performance with alternative CODECs, such as JPEG2000.

Reviewer #3 (Remarks to the Author):

The authors have submitted a revised version of their manuscript, and I greatly appreciate the effort they have invested in addressing all the referee comments and suggestions. I have carefully reviewed the resubmission.

The authors have responded thoroughly and thoughtfully to the feedback, including my own, and have made significant improvements. These revisions have notably enhanced both the clarity and rigor of the manuscript. In particular, the quantitative comparison now provides a much stronger foundation for the paper's conclusions.

Overall, I believe the authors have done an excellent job in revising their work, and I recommend the paper for publication.

Although we cannot publish your paper, it may be appropriate for another journal in the Nature Portfolio. If you wish to explore the journals and transfer your manuscript please use our manuscript transfer portal. You will not have to re-supply manuscript metadata and files, unless you wish to make modifications. For more information, please see our [manuscript transfer FAQ](http://www.nature.com/authors/author_resources/transfer_manuscripts.html?WT.mc_id=EMI_NPG_1511_AUTHORTRANSF&WT.ec_id=AUTHOR) page.

For Nature Portfolio general information and news for authors, see <http://npg.nature.com/authors>

Version 2:

Decision Letter:

**** Please ensure you delete the link to your author homepage in this e-mail if you wish to forward it to your co-authors. ****

Dear Dr Chen,

Your manuscript "Implicit Neural Image Field for Biological Microscopy Image Compression" has now been seen by 2 referees, whose comments are appended below. You will see that while they find your work of interest, they have raised points that need to be addressed before we can make a decision on publication. We apologize for the delay in getting a decision to you. However, there was some conflict regarding one of the reviewers' comments which took a while to resolve.

The referees' reports seem to be quite clear. Naturally, we will need you to address ***all*** of the points raised.

While we ask you to address all of the points raised, the following points need to be substantially worked on:

- To ensure real-world applicability, please validate performance on downstream tasks (e.g., DICE scores of segmentation on compressed data) or discuss this as a limitation.

Please use the following link to submit your revised manuscript and a point-by-point response to the referees' comments (which should be in a separate document to any cover letter):

Link Redacted

**** This url links to your confidential homepage and associated information about manuscripts you may have submitted or be reviewing for us. If you wish to forward this e-mail to co-authors, please delete this link to your homepage first. ****

To aid in the review process, we would appreciate it if you could also provide a copy of your manuscript files that indicates your revisions by making use of Track Changes or similar mark-up tools. Please also ensure that all correspondence is marked with your Nature Computational Science reference number in the subject line.

In addition, please make sure to upload a Word Document or LaTeX version of your text, to assist us in the editorial stage.

To improve transparency in authorship, we request that all authors identified as 'corresponding author' on published papers create and link their Open Researcher and Contributor Identifier (ORCID) with their account on the Manuscript Tracking System (MTS), prior to acceptance. ORCID helps the scientific community achieve unambiguous attribution of all scholarly contributions. You can create and link your ORCID from the home page of the MTS by clicking on 'Modify my Springer Nature account'. For more information please visit www.springernature.com/orcid.

We hope to receive your revised paper within three weeks. If you cannot send it within this time, please let us know.

Best regards,

Ananya Rastogi, PhD
Senior Editor
Nature Computational Science

Reviewers comments:

Reviewer #2 (Remarks to the Author):

Comparison experiments have been added in this version to enhance the overall quantification results, which is better than the previous version. Authors did address most prior concerns (e.g., contour consistency and more comparison methods), but one major point still remains unresolved.

Downstream validation (not only segmentation): Specifically, while the authors emphasize standard metrics (e.g., PSNR, SSIM) and perceptual guidance ablation (Fig. 6 in MainRevisedMS), Fig. 8 in ResponseToReferee highlights that these

metrics alone are insufficient to capture practical effectiveness of compression, particularly for microscopy images, where maintaining semantic fidelity (e.g., cell/nuclei segmentation accuracy) is crucial. To ensure real-world applicability, it is important to validate performance on downstream tasks (e.g., DICE scores of segmentation on compressed data). Authors are strongly encouraged to either (1) provide sufficient quantitative downstream task analyses (even using simple baselines) or (2) acknowledge this limitation and discuss its potential impact on biological applications in the revised paper.

Hope the authors can answer this key issue.

Reviewer #3 (Remarks to the Author):

The authors have submitted a novel version of their manuscript. Their responses have significantly improved the clarity and the overall scientific quality of the manuscript. I recommend accepting this work for publication.

Version 3:

Decision Letter:

Our ref: NATCOMPUTSCI-24-1168C

27th May 2025

Dear Dr. Chen,

Thank you for submitting your revised manuscript "Implicit Neural Image Field for Biological Microscopy Image Compression" (NATCOMPUTSCI-24-1168C). It has now been seen by the original referees and their comments are below. The reviewers find that the paper has improved in revision, and therefore we'll be happy in principle to publish it in Nature Computational Science, pending minor revisions to satisfy the referees' final requests and to comply with our editorial and formatting guidelines.

TRANSPARENT PEER REVIEW

Nature Computational Science offers a transparent peer review option for original research manuscripts. We encourage increased transparency in peer review by publishing the reviewer comments, author rebuttal letters and editorial decision letters if the authors agree. Such peer review material is made available as a supplementary peer review file. **Please remember to choose, using the manuscript system, whether or not you want to participate in transparent peer review.**

Please note: we allow redactions to authors' rebuttal and reviewer comments in the interest of confidentiality. If you are concerned about the release of confidential data, please let us know specifically what information you would like to have removed. Please note that we cannot incorporate redactions for any other reasons. Reviewer names will be published in the peer review files if the reviewer signed the comments to authors, or if reviewers explicitly agree to release their name. For more information, please refer to our <https://www.nature.com/documents/nr-transparent-peer-review.pdf> target="new">FAQ page.

Thank you again for your interest in Nature Computational Science. Please do not hesitate to contact me if you have any questions.

Sincerely,

Ananya Rastogi, PhD
Senior Editor
Nature Computational Science

ORCID

Reviewer #2 (Remarks to the Author):

Previous comments were mostly addressed.

Version 4:

Decision Letter:

Dear Dr Chen,

We are pleased to inform you that your Article "Implicit Neural Image Field for Biological Microscopy Image Compression" has now been accepted for publication in Nature Computational Science.

Once your manuscript is typeset, you will receive an email with a link to choose the appropriate publishing options for your paper and our Author Services team will be in touch regarding any additional information that may be required.

Authors may need to take specific actions to achieve compliance with funder and institutional open access

mandates. If your research is supported by a funder that requires immediate open access (e.g. according to [Plan S principles](https://www.springernature.com/gp/open-science/plan-s-compliance) or the [NIH public access policy](https://www.springernature.com/gp/open-science/us-federal-agency-compliance)) then you should select the gold OA route, and we will direct you to the compliant route where possible. Because authors warrant under our subscription licensing terms that they haven't committed to licensing any version of their article under a licence inconsistent with the terms of our agreement – including the applicable embargo period – publication under the subscription model isn't suitable for authors whose funders require no embargo.

Acceptance of your manuscript is conditional on all authors' agreement with our publication policies (see <https://www.nature.com/natcomputsci/for-authors>). In particular your manuscript must not be published elsewhere and there must be no announcement of the work to any media outlet until the publication date (the day on which it is uploaded onto our web site).

Before your manuscript is typeset, we will edit the text to ensure it is intelligible to our wide readership and conforms to house style. We look particularly carefully at the titles of all papers to ensure that they are relatively brief and understandable.

Once your manuscript is typeset, you will receive a link to your electronic proof via email with a request to make any corrections within 48 hours. If, when you receive your proof, you cannot meet this deadline, please inform us at rjsproduction@springernature.com immediately.

If you have queries at any point during the production process then please contact the production team at rjsproduction@springernature.com.

We welcome the submission of potential cover material (including a short caption of around 40 words) related to your manuscript; suggestions should be sent to Nature Computational Science as electronic files (the image should be 300 dpi at 210 x 297 mm in either TIFF or JPEG format). We also welcome suggestions for the Hero Image, which appears at the top of our [home page](http://www.nature.com/natcomputsci); these should be 72 dpi at 1400 x 400 pixels in JPEG format. Please note that such pictures should be selected more for their aesthetic appeal than for their scientific content, and that colour images work better than black and white or grayscale images. Please do not try to design a cover with the Nature Computational Science logo etc., and please do not submit composites of images related to your work. I am sure you will understand that we cannot make any promise as to whether any of your suggestions might be selected for the cover of the journal.

Best regards,

Ananya Rastogi, PhD
Senior Editor
Nature Computational Science

P.S. Click on the following link if you would like to recommend Nature Computational Science to your librarian: https://www.springernature.com/gp/librarians/recommend-to-your-library

** Visit the Springer Nature Editorial and Publishing website at www.springernature.com/editorial-and-publishing-jobs for more information about our career opportunities. If you have any questions please click here. **

Authors' Response to the Review of the Manuscript:

"Implicit Neural Image Field for Biological Microscopy Image Compression"

Manuscript ID: NATCOMPUTSCI-24-1168-T

Authors: Gaole Dai, Cheng-Ching Tseng, Qingpo Wu, Rongyu Zhang, Shaokang Wang, Ming Lu, Tiejun Huang, Yu Zhou, Ali Ata Tuz, Matthias Gunzer, Jianxu Chen, Shanghang Zhang

Dear Editors and Reviewers,

We sincerely thank the editorial team and all the reviewers for the valuable comments and constructive feedback on our manuscript. We have gone through all the comments carefully, addressed each Point raised, and made substantial revision accordingly. We believe these changes have significantly improved the manuscript.

Below, we summarize the major points raised by the editor and reviewers:

1. Qualitative and quantitative comparison with existing INR methods (Reviewer #1)
2. Expansion of the literature review (Reviewer #1)
3. Rectification of compressed image comparisons with different ratios (Reviewer #1)
4. Clarification of the difference between INIF and INR (Reviewer #2)
5. Preservation of essential information in residuals after network compression (Reviewer #2)
6. Robustness of neural networks in learning pixel values for different region sizes under transformations (Reviewer #2)
7. Sensitivity to learning rates compared to SIREN (Reviewer #2)
8. Comprehensive data comparison of compression times (Reviewer #2)
9. Speed comparisons with other techniques (Reviewer #3)
10. Evaluation of energy consumption and environmental impact (Reviewer #3)
11. Provision of quantitative metrics for all tests (Reviewer #3)

We have addressed each of these points in detail in this response letter. To further support the editors and reviewers in better understanding our work, we have included additional elements:

- (a) A ***For All the Reviewers and Editors*** 1 Section that summarizes and discusses all the raised questions concisely. This section includes most of the tables and figures referenced throughout our Points-by-Points response. We hope this section will help editors and reviewers gain a more comprehensive overview of INIF by integrating points from different reviewers.
- (b) We have used color-coding to highlight key information:
 - In all the tables, **best performances are in red** and **second-best performances are in blue**.
 - **Line numbers and modified words in the revised manuscript with tracked changes are indicated in brown**.
 - Quoted content from our manuscript or references is shown in gray.

We believe these additions will make the response letter easier to follow and more informative. We appreciate your time and effort in reviewing our work and look forward to your further feedback.

Contents

1 For All the Reviewers and Editors - Response for all the points in one story	3
1.1 What are the INR concept and key challenges faced by existing INR-based compressors?	3
1.2 How does INIF compare to other INR-based comparisons, especially in terms of "cost"?	5
1.2.1 Learned Optimizer & CODEC Prior Guidance minimizes Training costs . . .	6
1.2.2 Learned Optimizer minimizes Hyperparameter Tuning costs	7
1.2.3 Learned Optimizer & Application Appropriate Guidance minimizes Parti- tioning costs while Preserving Expressiveness	8
1.3 How does INIF provide a highly flexible and efficient decoding paradigm for the next-generation bioimage data?	13
1.4 Conclusion and Future Directions	14
2 For Reviewer #1 - Points 1, 2, 3	15
2.1 Comparison with existing implicit neural representation methods - Points 1	15
2.2 Expansion of literature review - Points 2	16
Learned Initialization	16
Cool-chic	17
C3	17
SCI	17
TINC	18
Comparison to INIF	18
2.3 Rectification of compressed image comparisons - Points 3	19
3 For Reviewer #2 - Points 4, 5, 6, 7, 8, 9	20
3.1 Difference between INIF and INR - Points 4	20
3.2 Preservation of essential information in residuals - Points 5	20
3.3 Robustness under transformations - Points 6	21
3.4 Sensitivity to learning rates - Points 7	23
3.5 Comprehensive data comparison of compression times (speed) with other techniques - Points 8 & 9	23
4 For Reviewer #3 - Points 9, 10, 11	24
4.1 Evaluation of energy consumption and environmental impact - Points 10	24
4.2 Provision of quantitative metrics for all tests - Points 11	24
5 Additional Points Raised by Reviewers	24

1 For All the Reviewers and Editors - Response for all the points in one story

We appreciate the thoughtful comments and suggestions from all the reviewers. In this section, we aim to provide a comprehensive overview of the supplementary experiments and explanations presented in our response.

1.1 What are the INR concept and key challenges faced by existing INR-based compressors?

INR represents a paradigm shift from conventional deep learning approaches that rely on explicit data input. The primary objective of INR is to learn a sample-wise function that transforms spatiotemporal coordinates into corresponding values using an artificial neural network (ANN) [1]. Typically, INR employs a multi-layer perceptron (MLP) [2], with the learning objective of minimizing the difference between the network's output and the ground truth (i.e., explicit data value). This flexible learning strategy allows INR to be deployed across various tasks, including 3D reconstruction [3, 4], registration [5], and super-resolution [6] (see Figure. 1).

The versatility of INR stems from its unique approach to data representation. While most neural networks are trained using explicit data values as input, applying non-linear transformations to produce complex manipulations of these values, INR aims to encode the data itself within the network parameters. This concept of learnable data parameterization dates back to the late 20th century, with classical works such as Hopfield networks [7] and Fourier networks [8, 9] exploring this approach. Our baseline comparison, SIREN [10], represents a recent advancement in leveraging modern ANNs for data representation. SIREN systematically analyzed how periodic activation functions (e.g., sine and cosine) enable neural networks to learn repetitive data structures more effectively.

However, SIREN's original work did not emphasize its potential for data compression. Subsequent INR compression studies have revealed some limitations of SIREN networks when constrained in size. For instance, SCI [11] and TINC [12] highlighted that a single parameter-limited SIREN network often struggles to simultaneously learn multiple frequency bands (e.g., high and low frequencies). To address this limitation, these works attempted to partition data into spectrally concentrated sub-blocks

Figure 1: Paradigm shift from learning how to transform explicit data values to learning how to represent data values implicitly.

Method	Cool-chic	C3	SCI	TINC	INIF
MAC	8.50E+04	8.00E+04	6.00E+04	5.00E+04	1.00E+05

and compress each sub-block individually. Other approaches, such as the Cool-chic series [13] and C3 [14] mentioned by Reviewer #1, have attempted to design INR-based compressors inspired by traditional CODEC methodologies. These methods incorporate intra-frame referencing, encoding and decoding data in an auto-regressive manner. While these approaches have demonstrated superior compression performance compared to SIREN, both theoretically and experimentally, we found they might face challenges in terms of compression and decompression efficiency.

Energy consumption comparison between state-of-the-art INR methods and INIF for compressing a 4GB dataset

Figure 2: Energy consumption and compression quality comparison to the state-of-the-art INR-based compression methods on large microscopy data. All the methods are tested on a single A100 40GB GPU with around 90% usage rate during training. INIF show 70-fold less energy usage compared to Cool-chic 3.1 but remains a comparable outcome.

1.2 How does INIF compare to other INR-based comparisons, especially in terms of "cost"?

As pointed out by Reviewer #2 in Points 8 and Reviewer #3 in Points 9, the efficiency of compression and decompression processes directly impacts the practical utility of a compression method. From an energy and environmental perspective, a compressor should aim to maximize the net benefit between the energy consumed during compression and the energy saved post-compression. However, evaluating the 'energy cost' is quite comprehensive, especially for INR-based methods, which are relatively new to the community. Based on our observations, recent works such as Cool-chic and C3 evaluated the Multiplication and Accumulation (MAC) metric during encoding and decoding. This is reasonable since, for CODECs, this metric provides a good approximation of the overall efficiency of the compressor. However, this is not the entire story for differentiable compressors, where iterative training is required. As we can see in Table. 1, INIF has a slightly higher MAC since we try to avoid partitioning the sample into sub-blocks and using fewer networks to increase convenience. However, our comparative experiments have revealed the actual high energy cost associated with excessive data partitioning and auto-regressive encoding approaches (see Figure 2). This high cost not only affects the practical applicability of these methods but also raises concerns about their overall energy efficiency and environmental impact. To delve deeper into the issues faced by INR-based compression methods and why relying solely on MAC is not rational anymore, we decomposed the cost during compression into several components:

(a) **Multiplication and Accumulation (MAC) per iteration:**

- Different compression methods may have vastly different computational complexity per iteration, directly impacting overall performance. The smaller MAC value reflects better basic efficiency of the algorithm for one iteration.

(b) **Iteration number per hyperparameter configuration:**

- For training-based methods (i.e. INR-based method), more iterations lead to longer total computation time, especially for large-scale data, the cost for one iteration would be even higher (because of using a larger MLP). For the non-trainable based method, the cost equals the per iteration MAC. This is also the main reason why CODEC can be much cheaper. Because they are undifferentiable and there is no need for training.

(c) **Hyperparameter search cost per Partition:**

- Except for INIF, other methods may require multiple trials to search for reasonable hyperparameters for each partition of the full data, affecting overall efficiency. For instance, we have searched for the Bitrate, Constant Rate Factor (CRF), and Adaptive Quantization (AQ) mode when using CODEC, learning rate, Group of Picture (GOP) strategies, for Cool-chic and C3.

(d) **Partition number per sample:**

- Most of the lossy compressors, including CODEC and the majority of neural compressors, are all designed for 2D-3D natural images. For High-dimensional biomedical data, using those methods may require partitioning strategies constituting a significant computational cost factor. For instance, in our 5D data test, we have to split the original data into 270 partitions when using a 3D compressor such as HEVC.

(e) **Decode cost:**

- Decoding speed is also important in certain application scenarios, especially for frequently accessed data. However, the major cost comes from the encoding part. Notice the decoding strategy depends on the usage of data, the flexibility and convenience of decoding should be the key factor when evaluating a general cost for decoding in practice.

For those reasons, we propose a comprehensive cost calculation formula:

$$\text{Encode cost} = \text{MAC} \times \text{Iter} \times \text{Hyper} \times \text{Part} \quad (1)$$

$$\text{Overall cost} = \text{Encode cost} + \text{Decode cost} \quad (2)$$

With this formula, we can reveal the true bottlenecks which limit the INR-based compressor’s efficiency and why we consider **using a learned optimizer could become an elegant solution for all these bottlenecks**. See detailed discussions below.

1.2.1 Learned Optimizer & CODEC Prior Guidance minimizes Training costs

The biggest bottleneck for all the INR-based methods is probably the cost when training iteratively, this is more severe when attempting to overfit the sample since it usually requires more iterative steps. The learned optimizer backbone we used is the Versatile Learned Optimizer (VeLO) [16]. From VeLO’s original technical report, Figure 1, the authors stated: "On 83 canonical tasks in the VeLOdrome benchmark, we train all the tasks faster than learning rate-tuned Adam. On about half of the tasks, we are more than 4x faster than learning rate-tuned Adam. On more than 14% of the tasks, we are more than 16x times faster." This dramatic reduction in iteration number motivates us to use VeLO for INR-based compression tasks further. We have also experimented to evaluate this reduction. Under the experimental setup in Table. 2, we have observed a much faster convergent speed when using the learned optimizer. Additionally, we also introduced the concept of CODEC’s prior guidance as an optional novel design. The main contribution of CODEC’s prior guidance in our INIF framework is to develop a mutual improvement pipeline by incorporating INIF as an adapter and equip CODEC with the ability to adjust their learning objective freely, meanwhile, CODEC can also boost the compression speed for INIF. (see Figure. 3). We believe this behaviour sheds light

Table 2: Experimental setup for dataset-level test using HiPSC multichannel 3D samples

Dataset	Image Size	Number of sample	Number of Cell Lines	Batch Size	Device
HiPSC [15]	~ (4, 65, 650, 920)	25	25	100000	Single RTX4090

Figure 3: Convergent speed comparison between different methods under compression ratio 256. INIF uses a learned optimizer to fasten the convergent speed, this can be further integrated with CODEC prior guidance to serve as a post-compression adapter with high efficiency.

Figure 4: Comparison of INIF and SIREN performance on stress test. Raw data (top left) and compressed results (top middle, right). The SIREN decompressed image demonstrates a case where inappropriate hyperparameters led to nonsensical results, highlighting the importance of proper parameter tuning in methods optimized by hand-crafted optimizers. The Loss curves for INIF and SIREN during training (bottom) further reveal the hidden cost parameter tuning. Note that even with nonsensical results, both methods show decreasing loss.

on a plausible solution for narrowing the gap between the iteration cost of CODEC and INR-based compressors.

1.2.2 Learned Optimizer minimizes Hyperparameter Tuning costs

Another cost which can be significantly reduced by a learned optimizer is the cost of searching hyperparameters. This cost is important in compressing tasks since we unavoidably need a set of hyperparameters for each sample no matter whether using CODEC or INR to get a close to optimal outcome. However, this cost is often ignored. For CODEC, a range of hyperparameters controls the compression ratio and affects the final quality. For example, we have searched for the Bitrate, Constant Rate Factor (CRF), and Adaptive Quantization (AQ) mode when using CODEC. For INR-based methods, the basic hyperparameter for all the baselines is the learning rate (LR) and additional hyperparameters for each method specifically e.g. GOP mode for Cool-chic, partition number for TINC etc. We have demonstrated improper LR would severely affect the final compression quality. This would be a bigger issue when compressing large data. For example, using the 4GB data in Figure 2. The result by SIREN with a default learning rate of 0.001 with cosine annealing scheduling is undesired (see Figure. 4). Nevertheless, without visualization or comparison to other baselines, we cannot have a clear judgment of the quality from the plotted loss curve of alone. In this case,

Figure 5: Comparison of raw microscopy images with HEVC and INIF compression results.

Figure 6: Compression performance comparison of various methods across different data dimensions. The graphs illustrate the relative costs of different methods over HEVC. INIF demonstrates consistent performance across dimensions, particularly excelling in high-dimensional scenarios.

the cost of searching a set of reasonable hyperparameters would be fairly high because this requires multiple search trails. INIF, in contrast, uses a learned optimizer to search for the optimal optimization direction and step size automatically. Which does not need to set a hard hyperparameter such as the LR before training. All the results shown are based on a single trial, this could be considered as a very convenient feature for further propagating INR-based compressors as a usable tool to the community.

1.2.3 Learned Optimizer & Application Appropriate Guidance minimizes Partitioning costs while Preserving Expressiveness

The final reason for using a learned optimizer is to preserve the unique attributes of INR to the maximum extent. As we mentioned earlier, most INR-related works, e.g., SIREN, are not designed

Table 3: Comparison of Compression Methods Across Different Data Dimensions

Data Dim	Metric	HEVC [17]	JPEG2000 [18]	SIREN (NIPS21) [10]	Cool-ctic (DCC24) [13]	C3 (CVPR24) [14]	TINC (CVPR23) [12]	INIF (w/o CODEC prior)
2D	MAC per Iteration	Low	Very Low	High	High	High	High	High
	Iteration per Hyperparameters	1	1	Mid	Mid	Mid	Mid	Mid
	Hyperparameters per Partition	Very Low	Very Low	Low	Low	Low	Low	Very Low
	Partition per Sample	1	1	1	1	1	1	1
3D	Decode	Mid	Mid	Mid	Low	Low	Mid	Mid
	MAC per Iteration	Low	Very Low	High	High	High	High	High
	Iteration per Hyperparameters	1	1	High	High	High	High	Mid
	Hyperparameters per Partition	Low	Mid	Low	Mid	Mid	Mid	Very Low
4D	Partition per Sample	1	High	Very Low	Mid	Mid	Mid	Very Low
	Decode	Mid	High	Mid	Low	Low	Mid	Mid
	MAC per Iteration	Low	Very Low	High	High	High	High	High
	Iteration per Hyperparameters	1	1	High	High	High	High	Mid
5D	Hyperparameters per Partition	High	Very High	Mid	High	High	High	Very Low
	Partition per Sample	High	Very High	Very Low	High	High	High	Very Low
	Decode	High	Very High	Mid	High	High	High	Mid
	MAC per Iteration	Low	Very Low	High	High	High	High	High
5D	Iteration per Hyperparameters	1	1	High	High	High	High	Mid
	Hyperparameters per Partition	Very High	Very High	High	Very High	Very High	Very High	Very Low
	Partition per Sample	Very High	Very High	Low	Very High	Very High	Very High	Low
	Decode	Very High	Very High	Mid	Very High	Very High	Very High	Mid

for compression, and the biggest challenge is the under-fitting issue when limiting the parameter size of the network. While we acknowledge the partition strategy adopted by most of the state-of-the-art INR compressors would be one of the solutions, the inconvenience during encoding and decoding in practical usage is also evident. Considering the data dimension and pixel depth of biomedical data often exceeds regular 2D, 3D, and 8-bit fashion. The issue along with partitioning becomes more severe. Under this setup, we discussed 3 compression scenarios. In the first scenario, when the data structure is suitable for CODEC and the compression quality of CODEC is acceptable, we recommend incorporating CODECs, as they are sufficient for most applications. Nevertheless, we enhance CODEC’s boundary by introducing residual compression with an INIF network to further improve compression quality. Our application-appropriate loss guidance serves as a post-CODEC adapter for specific user requirements that cannot be easily addressed by undifferentiable CODECs. Moreover, with biomedical images, the rest of the scenarios are very common and we believe INR-based methods have better potential to handle the challenges:

- (a) **Scenario 1: Low-dimensional data with unacceptable CODEC quality** In situations where CODEC quality is unacceptable, which occurs frequently in microscopy imaging (Figure 5), considering the speed advantage of CODEC becomes pointless, alternative compression strategies become necessary. INIF presents a viable option in these scenarios, offering a balance between compression quality and speed.
- (b) **Scenario 2: High-dimensional data** For data dimensions greater than 3, INIF unlike CODECs, neural CODECs, and INR-based methods heavily rely on partitions, it is designed to handle high-dimensional data natively. This capability is crucial in biomedical imaging, where high-dimensional, high-resolution, and high pixel-depth data are common.

To give a more direct comparison, we provided the cost of different methods under different data dimensions in Figure 6. INIF narrows the gap in terms of cost with HEVC by all the aforementioned advantages. We further summarized those observations in Table. 3.

Experimentally, we also see INIF hold a great balance between performance and network size without partitioning. Specifically, we have provided the quantitative results of all the experiments in our original manuscript in Table. 4. We have further performed two benchmark-level tests to validate the generalization ability of INIF:

- (a) We use the HiPCT dataset [19], which provides cellular-level imaging of several organisms at multiple anatomical levels. This dataset serves as the benchmark test for both SCI [11] and TINC [12] and they are representative state-of-the-art INR-based compressors for specialized biomedical data. All the preprocessing remains the same as TINC’s setup.
- (b) We also constructed our own benchmark based on the HiPSC dataset [15], the dataset information can be found in Table. 2.

From these experiments (see Table. 4, Table. 5, and Table. 6), we can see that INIF successfully overcomes the challenge of not partitioning. In most cases, we see INIF have comparable or even better results compared to other baselines. We hope this serves as a comprehensive response to Points 11 mentioned by Reviewer #3. However, we want to clarify that the reason we did not include all these

Table 4: Quantitative results comparison for all the experiments in the manuscript.

Experiment Figure #	Experiment goal	Experiment description	Target compression ratio	Metric	HEVC [17]	SIREN (NIPS21) [10]	INIF
2	Benchmark level test	3D, 16 bit, 25 samples	128	PSNR	76.25	76.8	78.3
				SSIM	0.9926	0.9965	0.998
			256	PSNR	75.75	76.5	78.2
				SSIM	0.992	0.9961	0.9976
3	Cross channel test	3D, 16 bit, 6 channels	256	PSNR	47.32	45.62	48.04
				SSIM	0.9832	0.9811	0.9853
4	High dimensional data test	5D, 8 bit	256	PSNR	71.02	79.89	81.21
				SSIM	0.9432	0.9822	0.9887
5	Application appropriate guidance	Segmentation downstream task	256	PSNR	73.15	73.7	74.79
				SSIM	0.9968	0.9921	0.9952
6		Low laser power (i.e. noisy) data	256	Dice	0.702	0.689	0.818
				PSNR	24.43	24.46	18.06
		Very low laser power (i.e. very noisy) data	256	SSIM	0.2445	0.242	0.051
				LPIPS	0.9871	0.9726	0.9167
				PSNR	75.77	79.37	75.19
				SSIM	0.9999	0.9999	0.9998
				LPIPS	0.7756	0.7625	0.7011
				PSNR	76.26	74.18	77.75
7	CODEC prior guidance	Speed up compression	256	SSIM	0.9999	0.9998	0.9999
				Energy (kWh)	0.0001	0.15	0.02

quantitative results in our original manuscript is not that we avoid comparing. The main reason is that we found using simple metrics such as PSNR or SSIM often contains bias and can be misleading when dealing with biomedical data (see Figure 7 right). For example, in the application-appropriate guidance test, we found that relying on PSNR or SSIM alone does not accurately reflect the actual performance in downstream tasks (such as the Dice score). Moreover, it is quite common to see biomedical images taken under specific microscope setups, e.g., varying laser power. When evaluating such data, naive metrics are not robust (see Figure. 7 left). This is also a very important topic often discussed by the biomedical imaging community. Recent publications have also pointed out that this phenomenon is widespread across different tasks, not only in compression [20, 21]. Consequently, we argue that a more nuanced approach to evaluating compression quality in biomedical imaging is necessary. To address this, we provide extensive visual comparisons throughout our manuscript, as these can often reveal qualitative differences that may not be captured by standard metrics alone. Also wherever applicable, we incorporate task-specific metrics (e.g., Dice coefficient for segmentation tasks) to evaluate the preservation of relevant features for particular applications. This multi-faceted approach to evaluation aligns with the complex nature of biomedical image data and the diverse requirements of the field. By combining quantitative metrics, visual comparisons, and application-specific evaluations, we aim to provide a comprehensive assessment of INIF’s performance across various biomedical imaging scenarios.

Besides evaluation, the flexibility of INIF’s learning objectives also inspired us to bring another novel design into INIF. Application-appropriate guidance allows for customized learning objectives tailored to specific downstream tasks or quality requirements, which showcases the potential of INR methods to adapt to the specific needs of scientific data compression. This customization capability is particularly valuable in the biomedical field, where the preservation of certain image features may be critical for subsequent analysis or interpretation.

Theoretically, we also aim to explore the potential of learned optimizers and application-specific guidance as a viable solution for mitigating underfitting issues without the need for partitioning. From an optimization perspective, the objective for any INR network can be reformulated as learning the function $f(\theta)$ parameterized by a neural network and mapping spatiotemporal coordinate space (x, y, z) (or any other implicit representation of data) to pixel intensity space (R, G, B) (or any other explicit data value). This mapping process can be formulated as:

$$f(x, y, z, \theta) = (R, G, B) \tag{3}$$

Table 6: Compression Performance Comparison on HiPSC Data. Note: For Cool-chic and C3, which do not support 16-bit depth, the original 16-bit images were converted to 8-bit for compression and then back to 16-bit for comparison. This highlights INIF’s superior capability in handling high-bit-depth biological data.

Ratio	Metric	JPEG2000 [18]	HEVC [17]	NeRF (ECCV20) [3]	SIREN (NIPS21) [10]	Learned Initializations (CVPR21) [26]	TINC (CVPR23) [12]	C3 (CVPR24) [14]	Cool-chic (DCC24) [13]	INIF
16	PSNR	30.3748	64.1356	56.5691	62.2605	60.8732	59.4859	42.62	46.89	67.4118
	SSIM	0.6622	0.9726	0.9612	0.9677	0.9628	0.9578	0.8513	0.8541	0.9866
	LPIPS	0.6219	0.0890	0.0988	0.0914	0.0955	0.0995	0.0795	0.0783	0.0801
32	PSNR	30.3003	63.3430	57.8868	61.4602	60.5188	59.5774	42.31	44.58	65.4842
	SSIM	0.6514	0.9701	0.9636	0.9653	0.9629	0.9605	0.8159	0.8188	0.9816
	LPIPS	0.6412	0.0912	0.0982	0.0934	0.0968	0.1001	0.0897	0.0885	0.0812
64	PSNR	30.8205	62.5066	58.8488	60.1666	60.5843	59.7489	41.01	42.28	63.8743
	SSIM	0.6792	0.9650	0.9506	0.9558	0.9466	0.9512	0.7789	0.7821	0.9812
	LPIPS	0.6201	0.0941	0.1030	0.0961	0.1060	0.0996	0.1003	0.0987	0.0882
128	PSNR	29.6318	61.5204	59.3098	59.7994	59.7470	59.6946	39.82	40.09	62.9531
	SSIM	0.6410	0.9629	0.9459	0.9537	0.9505	0.9473	0.7431	0.7459	0.9718
	LPIPS	0.6811	0.0972	0.1128	0.1056	0.1089	0.1121	0.1097	0.1085	0.0892
256	PSNR	29.2645	60.8283	59.1893	59.0253	59.0368	59.0138	34.55	37.82	62.3217
	SSIM	0.6367	0.9618	0.9601	0.9501	0.9539	0.9462	0.7057	0.7084	0.9634
	LPIPS	0.6909	0.0990	0.1019	0.1149	0.1144	0.1154	0.1197	0.1185	0.0978

Figure 7: Naive metrics are not always robust and even misleading when dealing with biomedical images.

where \mathcal{L} is a loss function measuring the difference between the predicted and true values, and C is a constant that limits the size or complexity of the network parameters θ . Both \mathcal{L} and C can be implemented in various ways, such as \mathcal{L} can be naively Mean Square Error (MSE) to target pixel level reconstruction, it can also be the application appropriate losses we proposed in the manuscript. C can be used to limit the number of parameters, apply weight quantization, or use regularization techniques. The introduction of parameter constraints also transforms our optimization problem into a search for Pareto Optimality. This further increases the complexity of the optimization process. The problem can be reframed as a multi-objective optimization:

$$\min_{\theta} (\mathcal{L}(f(x, y, z, \theta), (R, G, B)), \|\theta\|) \quad (5)$$

Figure 8: **Decoding INIF file in a pixel-wise fashion.** Unlike traditional file formats like TIFF that require complete decoding of the entire dataset, INIF enables pixel-wise decoding by providing the coordinates of the desired region of interest (ROI) to the INR network. INIF also supports multi-resolution decoding functionality and location-specific decoding even for irregularly shaped ROIs represented by binary masks.

where we seek to minimize both the reconstruction error and the parameter size simultaneously. This formulation explicitly highlights the trade-off between these two objectives, and the set of optimal solutions forms a Pareto front.

Traditional hand-crafted optimizers, which strictly adhere to gradient descent strategies, often struggle with this type of constrained optimization scenario. Once parameters reach a local optimum, it becomes challenging to make significant updates. However, dealing with constrained optimization typically requires continuous exchange between optima or even reparameterization. In this context, learned optimizers offer an alternative. They can, to some extent, ignore the limitations of gradients and make larger updates. A learned optimizer can be formulated as:

$$\theta_{t+1} = \theta_t + g(\nabla \mathcal{L}, \theta_t, \phi) \quad (6)$$

where g is a learned update rule parameterized by ϕ , unlike traditional optimizers that use fixed update rules, learned optimizers can adapt their behaviour based on the optimization landscape and the history of updates. This flexibility allows them to have the theoretical potential to escape local optima and explore the Pareto front more effectively.

1.3 How does INIF provide a highly flexible and efficient decoding paradigm for the next-generation bioimage data?

Another important advantage of INIF is that it is easier to apply to real-world applications due to its coherence, which allows for fast transfer and flexible decoding without over-partitioning. As in lines [881-902] in our manuscript, we Points out the decoding issues with the current data storing format and provide several useful decoding strategies that INIF can offer: "Commonly used file formats (e.g., JPG and PNG) can only use for regular 2D RGB images. Specialized file formats such as Ome-TIFF or Zarr have been developed for storing and decoding high-dimensional microscopy images [27, 28, 29]. Compared to Ome-Tiff, Zarr [27] offers more flexibility with additional features like multi-resolution storage, block-wise decoding, and parallel decoding using GPU. The evolution of these file formats indicates that the ultimate objective is to efficiently decode and visualize specific regions of interest (ROI) [30] in any microscopy data as needed since full decoding can be time-consuming and memory-intensive. INIF provides an excellent solution by enabling pixel-wise

decoding. Taking a 3D (XYZ) confocal microscopy image of DNA [31] as an example, assume our goal is to visualize slice 32. Decoding TIFF requires complete decoding of the entire dataset with dimensions X:924, Y:624, and Z:65 – including extracting all the pixel values followed by selecting only those belonging to slice 32 for visualization purposes. On the other hand, leveraging INIF’s pixel-wise decoding function allows us to simply provide the coordinates of slice 32 to the INR network for efficient decoding. Furthermore, by assigning a subset of coordinates, we can also support multi-resolution functionality – particularly useful when fast preview-based decodings are required without needing additional storage space for different resolution versions of the same data which is used in Zarr format. Additionally, even in special scenarios involving irregularly shaped ROIs represented by binary masks, we can still handle them through appropriate location-based decoding (see Figure. 8).". This would be very useful for biomedical data where they are frequently decoded for various downstream tasks.

Unlike in the domain of natural images, decompression of biomedical data rarely involves decoding the entire dataset at once. Most researchers prefer to first identify regions of interest (ROIs) using low-resolution images, and then perform full-resolution analysis specifically on these ROIs. However, the decoding methods adopted by most CODECs are more tailored to the demands of video streaming, employing frame-by-frame decoding. This approach led to the concept of Group of Pictures (GOP) to define I, P, and B frames. The disadvantage of this method is the inability to decode B or P frames independently. We also see recent INR works like Cool-chic and C3 have borrowed this auto-regressive-like decoding approach. While this method allows leveraging information from previous frames to the current frame, it also introduces a relatively rigid and slow decompression mode. Often, the information one wishes to access quickly requires waiting for the decompression of preceding information. For instance, when analyzing small cellular structures such as centrosomes, researchers may need to rapidly browse through the z-slices and identify several small regions within a large 3D dataset. Using traditional frame-based or auto-regressive decoding methods would require decoding all preceding data, even if only a small portion is of interest. This not only increases computational overhead but also significantly slows down the analysis process. INIF, on the other hand, offers a more flexible and efficient solution:

- (a) **Multi-scale Visualization:** INIF supports efficient multi-resolution functionality, enabling fast previews and progressive refinement of images without requiring additional storage for different resolution versions.
- (b) **Rapid ROI Decoding:** Researchers can quickly decode and analyze specific regions of interest at full resolution without processing the entire dataset. Even with irregularly shaped ROIs, INIF can still selectively decode only the relevant pixels, significantly reducing computational load.
- (c) **High-dimensional Analysis:** For Multi-channel time-lapse microscopy, INIF still supports decoding without partitioning into different sub-blocks. This avoids extra burden along with a complex file management system and would be useful for constructing a much more user-friendly storing format.
- (d) **Collaborative Research:** In scenarios where large datasets are shared among researchers, INIF’s flexible decoding allows each user to efficiently access only the parts of the data relevant to their specific research questions without any conflicts.

This flexibility makes INIF particularly well-suited for biomedical research applications where targeted, efficient data access is crucial.

1.4 Conclusion and Future Directions

In conclusion, our INIF method addresses several key challenges in INR-based compression, particularly for biomedical imaging data. By utilizing a learned optimizer, we overcome issues related to iterative training costs, hyperparameter tuning, and the need for data partitioning. Our comprehensive experiments demonstrate that INIF achieves competitive performance across various benchmark datasets and offers practical advantages in terms of energy efficiency and ease of use. Moreover, we highlight the importance of considering application-specific metrics, the limitations of conventional image quality measures when dealing with specialized data types like biomedical images, and how decoding paradigms might shift to be more flexible and convenient by using INIF. We hope our work

can pave the way for more efficient and adaptable compression techniques that can cater to the unique requirements of complex, high-dimensional biological microscopy image data.

2 For Reviewer #1 - Points 1, 2, 3

2.1 Comparison with existing implicit neural representation methods - Points 1

Figure 9: Energy consumption and compression quality comparison to the state-of-the-art INR-based compression methods on large microscopy data. All the methods are tested on a single A100 40GB GPU with around 90% usage rate during training. INIF show 70-fold less energy usage compared to Cool-chic 3.1 but remains a comparable outcome.

Table 7: Experimental setup for dataset-level test using HiPSC multichannel 3D samples

Dataset	Image Size	Number of sample	Number of Cell Lines	Batch Size	Device
HiPSC [15]	~ (4, 65, 650, 920)	25	25	100000	Single RTX4090

We appreciate the reviewer’s suggestion to include comparisons with existing methods. In response, we have significantly expanded our experimental comparisons, including additional baselines from both traditional CODECs and recent neural network-based approaches. While recent state-of-the-art

INR methods like Cool-chic [13] and C3 [14] have demonstrated impressive results, they encounter significant challenges when dealing with high-dimensional data, particularly in 4D and 5D scenarios with high pixel depth such as 16-bit. In terms of performance, when the data dimension and pixel depth meet Cool-chic/C3’s requirements (2D-3D, 8-bit data), INIF can exhibit comparable or superior decompressed quality. However, in terms of efficiency, INIF is 50-70 times more efficient (see Figure. 9). Based on our observations, the issues with Cool-chic, C3, and several other state-of-the-art INR compressors such as SCI [11] and TINC [12] stem from increased partition complexity. Consequently, this leads to significant encoding costs for compressing sub-blocks individually and also incurs additional expenses such as extensive hyperparameter tuning requirements; moreover, it even results in higher decoding complexity for higher-dimensional data if decoded auto-regressively. In contrast, our proposed INIF method maintains consistent performance across all the dimensions by reducing the inconvenience caused by over-partitioning while eliminating the need for hyperparameter tuning all through a learned optimizer approach. We have included a benchmark-level comparison test using the configurations listed in Table 7. Under this setup, we conducted extensive comparisons with state-of-the-art methods such as TINC, Cool-chic, and C3. The results presented in Table 6, demonstrate that INIF consistently outperforms these methods across various metrics.

2.2 Expansion of literature review - Points 2

We appreciate this suggestion for the expansion of the literature review. In the Section "**For All the Reviewers and Editors**" 1.2 of this response letter, we provided a detailed explanation of the core concept of INR and the key challenges faced by other existing INR-based compression methods. We have included this extra clarification information in the revised manuscript, which can be found on lines [55-76] of the revised manuscript. To ease the reviewers, we also pasted the updated paragraph here:

"Briefly, INR learns a sample-specific function using an artificial neural network (ANN), typically a multi-layer perceptron (MLP) [32], to map multi-dimensional spatiotemporal coordinates of any data to its corresponding pixel values. The core concept of INR focuses on representing data values through the parameter values of the neural network. On one hand, works such as SIREN [33] systematically analyze how periodic activation functions (e.g., sine and cosine) enable neural networks to learn repetitive data structures more effectively. On the other hand, the flexibility of INR has inspired researchers to deploy this learning strategy across various applications, including 3D reconstruction [34, 4], registration [5], and super-resolution [6]. Recent advances have also uncovered the potential of INR-based compressors [35, 36, 13, 14, 11, 12]. By limiting the parameter size of the ANN, supported with dimensional information, INR-based compressors theoretically enable controllable high compression ratios and on-demand reconstruction of the full data or any specific regions (even disconnected) to arbitrary data with a unified network. However, due to network size limitations, capturing intricate high-frequency details poses a significant challenge for INR networks [37]. Recent INR-based compressors have attempted to borrow partition strategies from traditional CODECs, transforming data into smaller sub-blocks for individual compression. These methods [13, 14, 11, 12] reported better compression quality. Still, the drawback of such sub-block-specific learning is that it might require considerable time to compress a single image. At the same time, performance still largely depends on tuning certain training hyperparameters, e.g., the learning rate for each sub-block. This issue is particularly pronounced for large bioimages."

Further details: To facilitate the reviewing process, we provide expanded details of other related methods and how they are different from our proposed INIF method.

Learned Initialization

Tancik et al. [26] introduced a meta-learning method to optimize the initial weights of coordinate-based neural networks. The primary innovation of this approach lies in its ability to learn a general initialization that can accelerate the optimization process for individual signals.

Key Innovations:

- Meta-learning framework for weight initialization
- Faster convergence for single signal optimization
- Improved generalization across a distribution of signals

Limitations compared to INIF:

- **High Cost of Meta-learning:** Although learning the initialization does not require the network to be trained very sufficiently. This process still needs to loop through and compress the dataset. Balancing the extra cost for the initial learning process and the reduction in cost within the formal compression process would be crucial.
- **Dataset Requirements:** Learned initialization requires a dataset of similar samples to learn an effective initialization. This is often not feasible in biological microscopy, where private, independent samples are also prevalent. The method may fail when compressing a single unique sample or when samples within a dataset are not closely related.

Cool-chic

Ladune et al. [13] improved upon the COIN [35] method by introducing multi-scale features and block-based encoding/decoding strategies. The main innovation of Cool-chic lies in combining the efficiency of traditional CODECs with the expressiveness of neural networks.

Key Innovations:

- Integration of multi-scale features
- Block-based encoding and decoding
- Hybrid approach combining traditional CODECs and neural networks

Limitations compared to INIF:

- **Data Format Restrictions:** Cool-chic primarily targets 2D and 3D data with 8 or 10-bit depth, adopting traditional CODEC GOP (Group of Pictures) strategies, it faces challenges when dealing with high-dimensional (4D, 5D) or high-bit-depth (16-bit) data, which are common in biological microscopy. This design limits its application to higher-dimensional or higher bit-depth data, losing the advantage of INR's flexible data representation.
- **High Cost of Auto-regressive** Cool-chic encodes sub-blocks of data similar to CODEC in an auto-regressive fashion, this could reduce the per-pixel MAC. However, it lacks support for parallel compression of each sub-block, causing the actual compression cost significantly higher. From our experiment in Figure 9, it is 70-fold slower than INIF when compressing, this phenomenon would be even more significant for multi-dimensional data that doesn't follow traditional video structures.

C3

Kim et al. [14] further improved upon Cool-chic, enhancing compression performance while maintaining low decoding complexity. C3's innovations include improved optimization strategies and more flexible architecture choices.

Key Innovations:

- Enhanced optimization techniques
- Flexible architecture adaptation
- Improved rate-distortion performance

Limitations compared to INIF:

- **Data Format, Auto-regressive and Partition Issue similar to Cool-chic** Although C3 further reduce the complexity of Cool-chic and reaches a lower per-pixel MAC, the core issue remains similar to Cool-chic. We found it is still around 60-fold slower than INIF and with lower decompressed quality (see Figure 9).

SCI

Yang et al. [11] proposed SCI, which focuses on adapting INR-based compression to data with broad spectrum distributions, particularly in biomedical imaging.

Key Innovations:

- Adaptive partitioning strategy based on spectrum concentration
- Funnel-shaped neural network architecture for efficient block-wise compression
- Parameter allocation strategy based on spectral characteristics

Limitations compared to INIF:

- **Computational Efficiency:** While SCI improves compression quality, it may require longer compression times due to its adaptive partitioning and block-wise compression approach.
- **Hyperparameter Sensitivity:** The performance of SCI can be sensitive to hyperparameter choices, which may require tuning for different types of data.

TINC

Yang et al. [12] introduced TINC as an improvement of C3, which organizes multiple INRs in a tree structure to capture both local and non-local redundancies in the data.

Key Innovations:

- Tree-structured organization of multiple INRs
- Hierarchical parameter sharing mechanism
- Improved compression for complex data with spatial varying feature distribution

Limitations compared to INIF:

- **Memory Complexity:** The tree structure, while reducing the time complexity of SCI, introduces additional memory complexity in both compression and decompression processes. This would be more significant when facing large data.
- **Block Artifacts:** TINC partition the sample into sub-blocks and compresses each block with separate neural networks with shared weight. However, when sharing the weight of different neural networks, artifacts in one sub-block can also replicated to other sub-blocks, bringing block artifacts.

Comparison to INIF

In contrast to these methods, our proposed INIF approach offers several unique advantages for compressing biological microscopy data:

- (a) **Compression-Specific Challenges:** INIF tackles the unique challenge of underfitting in compression tasks. When the network needs to represent a significantly larger data space within the constraints of limited file size, To address this limitation, most works attempted to partition data into spectral concentrated sub-blocks and compress each sub-block individually. However, we found they might face challenges in terms of compression and decompression efficiency, particularly in the context of biological microscopy data which is both high-dimension and large in size. INIF’s design specifically addresses the aforementioned issues without the inconvenience and cost burden of over-partitioning, enabling effective compression of complex, high-dimensional data.
- (b) **Learned Optimizer:** As a brand new path to overcome stagnation during gradient descent, INIF introduces a learned optimizer [38]. This innovative approach predicts gradient directions and optimizes the network, resulting in faster convergence and better implicit representation of the data as an entirety. Essentially, the network learns how to efficiently overfit the data, a crucial ability in compression tasks.
- (c) **Comprehensive Workflow:** INIF is not just a representation method but a complete compression pipeline. It incorporates optional designs at various stages:
 - Pre-compression: CODEC prior guidance
 - In-compression: Application-appropriate guidance
 - Post-compression: ROI (Region of Interest) and multi-resolution decoding

These components are seamlessly integrated into INIF's core design without incurring additional computational burdens, enhancing its practical applicability.

- (d) **Adaptability and Flexibility:** INIF is highly adaptable to specific compression requirements. Its flexibility allows it to handle various data types and dimensions common in biological microscopy, making it particularly suitable for scientific data compression.
- (e) **Competitive Performance:** The combination of these features enables INIF to compete effectively with commercial CODECs in practical applications, especially in scenarios where traditional methods struggle, such as with high-dimensional or variable-resolution data.

In conclusion, while Learned Initialization, Cool-chic, C3, SCI, and TINC have made significant contributions to the field of neural compression, INIF demonstrates unique advantages in handling complex biological microscopy data. INIF's design better accommodates the diversity and complexity of biological microscopy data while effectively reducing computational costs through innovative optimization strategies and a hybrid approach. This positions INIF as a highly promising solution in the field of biological microscopy data compression, addressing key limitations of existing methods and opening new possibilities for efficient data management in scientific research.

2.3 Rectification of compressed image comparisons - Points 3

We appreciate the reviewer's concern regarding the fairness of comparisons, particularly with HEVC. (Another concern is the fairness of comparisons to HEVC, especially in Figure. 2. Although HEVC can be challenging to control for fixed compression ratios, comparing compressed images with significantly different ratios is not ideal. This discrepancy makes it difficult for readers to fully appreciate the impact of the proposed method.) We have addressed this issue in several ways:

- (a) As mentioned in our manuscript lines [148-153]: "Even with meticulous adjustment of hyper-parameters such as bitrate [39], constant rate factor (CRF) [40], and adaptive quantization (AQ) mode [41], it is unable to control the compression rate throughout the HEVC experiment. In contrast, our proposed INIF method consistently maintains clear visibility of cellular contours and retains considerable textural details under controllable high compression ratios." , one of the major drawbacks of traditional CODECs is the difficulty in precisely controlling compression ratios. To be specific, the "meticulous adjustment" we conducted is a hyperparameter sweep for our CODEC competitors, including Bitrate, Constant Rate Factor (CRF), and Adaptive Quantization (AQ) mode, with a search depth of 10. Bitrate directly controls the output file size, CRF balances compression efficiency and video quality, and AQ-mode adjusts quantization strength based on frame complexity. The combination of these parameters allows for fine-grained control over the compression process, theoretically enabling ratio management to some extent. We performed this grid search process for each sample to find a reasonable parameter combination. However, despite this extensive parameter tuning, there are still scenarios where precise control of compression ratios remains challenging for traditional CODECs. We deliberately included such cases in our study to highlight this limitation. For instance, in Figures 2 and 6 of our manuscript, we showcase examples where HEVC struggles to achieve the target compression ratio, even after our thorough hyperparameter optimization. These examples underscore the inherent variability in CODEC performance across different types of microscopy data.
- (b) In most cases, we have successfully matched the compression ratios between INIF and other methods. This is evident in Figures 3, 4, 5, 6, and 7 of the main text, where comparisons are made at equivalent compression ratios.
- (c) In Figure 4, we demonstrate that INIF achieves better results than HEVC on high-dimensional data compression tasks while using 4-fold less bitrate.
- (d) In cases where exact ratio matching was not possible (e.g., Figures 2 and 6), we emphasize that HEVC can face severe failures with certain types of data, such as noisy images or images with drastic changes between frames/depth. In contrast, INIF, rooted in implicit neural representations, offers a more robust compression approach for diverse data types.

For your convenience to compare, we have listed the quantitative results of the aforementioned Figures in Table. 4.

3 For Reviewer #2 - Points 4, 5, 6, 7, 8, 9

3.1 Difference between INIF and INR - Points 4

In the Section "For All the Reviewers and Editors" 1.1 of this response letter, we provided a detailed explanation of the core concept of INR and the key challenges faced by other existing INR-based compression methods. In short, INR represents a paradigm shift from conventional deep learning approaches that rely on explicit data input to implicit representation learning. INIF, in contrast, is a complete workflow specifically designed for compression tasks, leveraging INR as its core learning strategy, offering a new solution that bridges the gap between the theoretical strengths of INR and the practical demands of data compression in fields like biological microscopy. We have included this extra clarification information in the revised manuscript, which can be found on lines [55-76] of the revised manuscript. To ease the reviewers, we also pasted the updated paragraph here:

"Briefly, INR learns a sample-specific function using an artificial neural network (ANN), typically a multi-layer perceptron (MLP) [32], to map multi-dimensional spatiotemporal coordinates of any data to its corresponding pixel values. The core concept of INR focuses on representing data values through the parameter values of the neural network. On one hand, works such as SIREN [33] systematically analyze how periodic activation functions (e.g., sine and cosine) enable neural networks to learn repetitive data structures more effectively. On the other hand, the flexibility of INR has inspired researchers to deploy this learning strategy across various applications, including 3D reconstruction [34, 4], registration [5], and super-resolution [6]. Recent advances have also uncovered the potential of INR-based compressors [35, 36, 13, 14, 11, 12]. By limiting the parameter size of the ANN, supported with dimensional information, INR-based compressors theoretically enable controllable high compression ratios and on-demand reconstruction of the full data or any specific regions (even disconnected) to arbitrary data with a unified network. However, due to network size limitations, capturing intricate high-frequency details poses a significant challenge for INR networks [37]. Recent INR-based compressors have attempted to borrow partition strategies from traditional CODECs, transforming data into smaller sub-blocks for individual compression. These methods [13, 14, 11, 12] reported better compression quality. Still, the drawback of such sub-block-specific learning is that it might require considerable time to compress a single image. At the same time, performance still largely depends on tuning certain training hyperparameters, e.g., the learning rate for each sub-block. This issue is particularly pronounced for large bioimages."

3.2 Preservation of essential information in residuals - Points 5

We appreciate the reviewer's concern about information preservation. We have added a detailed explanation of our approach to this issue (lines [262-277] in the revised manuscript):

"Specifically, in this mode, the initial compression attempt is performed using any CODECs such as HEVC. Typically, this approach is cost-effective as these CODECs are untrainable and require only a few iterations to complete the compression process. However, it also implies that the adopted CODECs here are less adaptable and lack means for further optimizing the quality of decompressed images. In order to address this issue, we transform our INIF backbone into an adapter [42] by substituting the raw image ground truth with residuals obtained by subtracting CODEC-decompressed images from raw images. By conducting a short additional training phase using INIF, we can compare the adapted result with the original result to determine if overwriting the current slice is necessary. It should be noted that in this case, the evaluation standard is also adaptive; in our experiment below, we selected centrosome count as an application-specific metric. For more general usage scenarios, selection based on common metrics such as PSNR or SSIM would also be feasible. With this straightforward design approach, we not only enable adjustable compression objectivity to reduce blocky artifacts associated with CODEC but also expedite the training process of INIF by reducing information load."

The main contribution of CODEC's prior guidance in our INIF framework is to develop a mutual improvement pipeline by incorporating INIF as an adapter and equipt CODEC with the ability to adjust their learning objective, meanwhile, CODEC can also boost the compression speed for INIF. We have validated this by conducting a benchmark-level test using the dataset listed in Table. 2. From the result in Figure. 3, HEVC as a powerful commercial used CODEC can already outperform most of the baseline with lower cost. However, INIF expands the boundary of CODEC's ability by incorporating the aforementioned design. It's important to note that INIF is not a lossless compression method, and

we deliberately avoid using the term "ensure" as some essential information in the residuals may be inevitably ignored during compression. Our approach to maximizing the preservation of essential information is as follows:

- (a) **Hybrid Compression Strategy:** We employ a two-stage compression process involving both traditional CODEC and INIF. This hybrid approach allows us to leverage the strengths of both methods.
- (b) **Quality Evaluation Metric:** After the complete compression by both CODEC and INIF, we evaluate the decompressed quality using a mixed metric Q_{mix} that combines PSNR, SSIM, and LPIPS. The calculation is formulated as:

$$Q_{mix} = \text{NORM}(\text{PSNR} + \text{LPIPS} + (1 - \text{SSIM})) \quad (7)$$

This composite metric provides a more comprehensive assessment of image quality, capturing both structural similarity and perceptual aspects.

- (c) **Slice-wise Evaluation:** We calculate the Q_{mix} metric slice-wise for both pre-INIF (CODEC only) and post-INIF (CODEC + INIF) results. This granular approach allows us to assess the quality improvements at a more detailed level.
- (d) **"Winner-Take-All" Selection:** We implement a "winner-take-all" selection strategy for each slice. The decoding strategy (whether to add the learned residual from INIF or not) is determined based on the higher Q_{mix} score for each slice.

This approach offers several advantages:

- (a) **Adaptive Compression:** By selecting the better result for each slice, we ensure that the compression adapts to local image characteristics.
- (b) **Quality Optimization:** The method maximizes the overall quality of the compressed data by choosing the best representation for each part of the image.
- (c) **Balancing Efficiency and Quality:** It allows us to benefit from the efficiency of CODEC compression while still leveraging the advantages of INIF where it provides significant improvements.

While this method does not guarantee lossless compression, it represents a thoughtful approach to balancing compression efficiency with the preservation of essential information. By combining traditional CODEC techniques with our novel INIF approach and employing a sophisticated quality assessment and selection process, we strive to achieve optimal results in terms of both compression ratio and image quality preservation.

3.3 Robustness under transformations - Points 6

We appreciate the reviewer’s insightful question regarding the need for standardization before compressing unregistered data with affine and deformable transformations. We are pleased to clarify that no such standardization is required, thanks to the inherent robustness of INIF. We have conducted additional experiments and analysis to address this concern:

From a mathematical perspective, INIF learns the function $f(\theta)$ parameterized by a neural network to project coordinate space (x, y, z) to pixel intensity space (R, G, B) (assuming 3D colour data compression). This projection process can be formulated as:

$$f(x, y, z, \theta) = (R, G, B) \quad (8)$$

When the original data undergoes affine or deformable transformation, the current coordinate space (x, y, z) becomes a resampled outcome of the original coordinate space (x_o, y_o, z_o) . We can express this transformation as:

$$(x, y, z) = T(x_o, y_o, z_o) \quad (9)$$

Figure 10: Compression of multiple unregistered datasets with a single INIF model. Our experimental setup generates 8 different unregistered data by randomly distorting the raw data from different angles. The compression goal is to compress these 8 samples into 1 INIF.

Where T represents the transformation function. In this case, the INIF learning process can be described as:

$$f(T(x_o, y_o, z_o), \theta) = (R, G, B) \quad (10)$$

This formulation demonstrates that INIF inherently learns the mapping from the transformed coordinate space to the pixel intensity space. The registration status of the data does not fundamentally change the nature of this mapping process, as there still exists a one-to-one correspondence between the transformed coordinates and their respective pixel intensities.

To further demonstrate the robustness of INIF, we conducted experiments in an extreme scenario:

- (a) We trained a single INIF model to simultaneously compress multiple unregistered datasets.
- (b) In this scenario, a single coordinate potentially corresponds to multiple pixel values, as each unregistered dataset has its own transformation parameters.

Mathematically, this scenario can be represented as:

$$f(T(x_o, y_o, z_o), \theta) = (R_i, G_i, B_i), \quad i = 1, 2, \dots, n \quad (11)$$

Where i denotes different unregistered datasets, each with its transformation T and corresponding pixel values (R_i, G_i, B_i) .

Interestingly, our experiments (see Figure. 10) revealed that even in this challenging scenario, INIF was able to learn a unified representation to a certain extent. This ability to capture and compress multiple unregistered datasets simultaneously is a unique advantage of INIF that cannot be achieved through traditional CODEC methods.

The success of INIF in handling both single unregistered datasets and multiple unregistered datasets can be attributed to its fundamental approach to learning a continuous mapping function. This function

is flexible enough to accommodate various spatial transformations and even multiple correspondences between spatial coordinates and pixel intensities.

In conclusion, INIF demonstrates remarkable robustness to data transformations, eliminating the need for pre-registration or standardization. This feature not only simplifies the compression pipeline but also extends the applicability of INIF to scenarios where data alignment is challenging or impractical.

3.4 Sensitivity to learning rates - Points 7

One of the major advantages of our learned optimizer in INIF is its independence from the learning rate (also refer to the "**For All the Reviewers and Editors**" 1.2.2 for full details about the advantages of the learned optimizer in INIF). This characteristic is consistently emphasized throughout our manuscript ("lines [93-97]: The raw image serves as ground truth for calibrating the similarity between the reconstructed image and the raw image with a similarity loss (e.g. mean square error), subsequently optimized by a learned optimizer [43] (i.e., no need to set learning ratio or other optimization hyper-parameters). ") and visually demonstrated in Figures 2, 3, and 4. Unlike other INR-based methods that rely on hand-designed optimizers, such as Adam, and struggle to find an optimal learning rate, INIF remains unaffected by this issue.

This feature proves highly valuable in practical scenarios, particularly when compressing large files where each training trial incurs significant computational costs. The need for multiple trials to search for optimal hyperparameters, especially the learning rate, further amplifies this concern. While most studies in the literature focus on comparing efficiency using a relatively optimal learning rate, they often overlook the substantial cost associated with finding such hyperparameters. In contrast, INIF places greater emphasis on practical efficiency.

To illustrate this Points, we provide a realistic example:

- (a) The file size of the raw data in Figure 4 (top) is approximately 4 GB.
- (b) Each training trial takes around 1.5 hours with a compression ratio set to 512.
- (c) As shown in the loss curve in Figure 4 (bottom), while SIREN experiences a period of rapid loss decrease and appears to converge, the actual compressed quality cannot be evaluated without visualization support.

This scenario necessitates several additional trials to confirm the appropriate range of hyperparameters, incurring significant hidden costs. These costs are often overlooked in the research community but are crucial in practical applications. For instance, to avoid producing nonsensical results as shown in Figure 4 (top right), multiple training iterations with different hyperparameters may be required.

INIF addresses these challenges by eliminating the need for extensive hyperparameter tuning, particularly for the learning rate. This not only saves computational resources but also makes the compression process more robust and reliable. The learned optimizer in INIF adapts to the specific characteristics of the data being compressed, automatically adjusting its behaviour without manual intervention.

In summary, while INIF's learned optimizer may not always produce the global optimal conditions, its practical advantages in terms of ease of use, robustness, and elimination of costly hyperparameter searches make it a superior choice for real-world applications, especially when dealing with large-scale, diverse datasets typical in biological microscopy.

3.5 Comprehensive data comparison of compression times (speed) with other techniques - Points 8 & 9

Please refer to the "**For All the Reviewers and Editors**" 1.2 Section regarding the extended discussion and quantitative comparison between INIF and other techniques in terms of "cost" (the part about compression speed).

4 For Reviewer #3 - Points 9, 10, 11

4.1 Evaluation of energy consumption and environmental impact - Points 10

We thank the reviewer for raising this important Point regarding energy consumption and environmental impact. Please refer to the "**For All the Reviewers and Editors**" 1.2 Section regarding the extended discussion and quantitative comparison between INIF and other techniques in terms of "cost" (the part about energy consumption), specifically Table 3 and Figure 2. In short, while the initial compression process of INIF may be more computationally intensive than some traditional methods, its energy efficiency in handling high-dimensional data and the long-term benefits in storage and data transfer make it an environmentally friendly choice for large-scale scientific data compression.

4.2 Provision of quantitative metrics for all tests - Points 11

We appreciate the reviewer's request for comprehensive quantitative metrics. In response, we have conducted extensive experiments and compiled detailed quantitative results for all our tests. Please refer to the "**For All the Reviewers and Editors**" 1.2 Section, especially Table 4 summarizing the key experiments and metrics used in our study, and the respective results in Table 5 and Table 6. In short, we see INIF have comparable or even better results compared to other baselines. However, we want to clarify that the reason we did not include all these quantitative results in our original manuscript is not to avoid quantitative comparison. The main reason is that we found using simple metrics such as PSNR or SSIM often contains bias and can be misleading when dealing with biomedical data (see Figure 7 right). For example, in the application-appropriate guidance test, we found that relying on PSNR or SSIM alone does not accurately reflect the actual performance in downstream tasks (such as the Dice score). Moreover, it is quite common to see biomedical images taken under specific microscope setups, e.g., varying laser power. When evaluating such data, naive metrics are not robust (see Figure 7 left). This is also a critical topic often discussed by the biomedical imaging community. Recent publications have also pointed out that this phenomenon is widespread across different tasks, not only in compression [20, 21]. Consequently, we argue that a more nuanced approach to evaluating compression quality in biomedical imaging is necessary. Our strategy involves:

- **Visual comparisons:** We provide extensive visual comparisons in our figures, as these can often reveal qualitative differences that may not be captured by standard metrics alone.
- **Application-appropriate guidance:** We introduce the concept of application-appropriate guidance, which allows for customized learning objectives tailored to specific downstream tasks or quality requirements.
- **Task-specific metrics:** Where applicable, we incorporate task-specific metrics (e.g., Dice coefficient for segmentation tasks) to evaluate the preservation of relevant features for particular applications.

This multi-faceted approach to evaluation aligns with the complex nature of biomedical image data and the diverse requirements of the field. By combining quantitative metrics, visual comparisons, and application-specific evaluations, we aim to provide a comprehensive assessment of INIF's performance across various biomedical imaging scenarios.

Furthermore, the flexibility of INIF's learning objectives through application-appropriate guidance showcases the potential of INR methods to adapt to the specific needs of scientific data compression. This customization capability is particularly valuable in the biomedical field, where the preservation of certain image features may be critical for subsequent analysis or interpretation.

5 Additional Points Raised by Reviewers

Comment from Reviewer 1: How is the compression ratio calculated? Does it include the network configuration, network weights, and latent code? The compression ratio is calculated by directly comparing the size of the raw image with the total size of all the files, including network weights and network configuration (metadata) required for transfer, storage, and decoding. This direct calculation accurately reflects real-world scenarios where a specific compression ratio is desired for raw data. It should be noted that we do not utilize latent code as input, unlike previous works such as

Cool-chic [13], C3 [14], and NerV [25]. This implies that we can freely generate input coordinates from the metadata of the raw data shape based on demand. Consequently, this approach reduces the necessity to transmit explicit latent codes and enables more controllable compression ratios with flexible decoding strategies.

Comment from Reviewer 1: Are we training an INIF for each image stack? What are the preprocessing steps for the raw images to be compressed? The training process involves creating an INIF for each image stack, showcasing the ability of this method to handle image stacks with varying dimensions. This unique advantage sets it apart from most other methods. The preprocessing steps before compression are outlined in lines [547-554] in the manuscript: " The initial step involves normalizing the input images to ensure consistency in their range. We have implemented Min-Max normalization [44], which entails defining the minimum and maximum intensity values and subsequently mapping all the values linearly within this range. Specifically, we have set the intensity range from 0 to 100. Before training, another preprocessing step is performed wherein we derive coordinates by extracting shape metadata from the target data and initializing coordinates with identical shapes corresponding to the data. Furthermore, these coordinates are normalized within each dimension to a range of -1 to 1. " These are all the necessary steps for the data before compression.

Comment from Reviewer 3: I recommend that the authors provide complementary work on the Points spread function of the imaging system in Figure 7. We totally agree that additional imaging information is important complementary information. For most of the experiments in this work, we used only public data from prior biological publications, except for the stress test on very large data. For the public data, we don't have any additional information besides the details provided by the original publication. For example, all the necessary image acquisition details of the HiPSC dataset can be found in the method Section and supplementary methods of the original publication [15]. We added important acquisition details in the supplementary material (Appendix C) for the internal data we used for a stress test.

Comment from Reviewer 3: The size of the images in Figures 2, 3, 4, and 6 should be increased, as their differences are often difficult to discern. We increased the size of the images of the corresponding pictures, please refer to the Figures below

Figure 11: Increased Size Figure 2.

Figure 12: Increased Size Figure 3.

Figure 13: Increased Size Figure 4.

Figure 14: Increased Size Figure 6.

References

- [1] Oludare Isaac Abiodun, Aman Jantan, Abiodun Esther Omolara, Kemi Victoria Dada, Nachaat AbdElatif Mohamed, and Humaira Arshad. State-of-the-art in artificial neural network applications: A survey. *Heliyon*, 4(11), 2018.
- [2] Hind Taud and Jean-Francois Mas. Multilayer perceptron (mlp). *Geomatic approaches for modeling land change scenarios*, pages 451–455, 2018.
- [3] Ben Mildenhall, Pratul P Srinivasan, Matthew Tancik, Jonathan T Barron, Ravi Ramamoorthi, and Ren Ng. Nerf: Representing scenes as neural radiance fields for view synthesis. In *European Conference on Computer Vision*, pages 405–421. Springer, 2020.
- [4] Jeong Joon Park, Peter Florence, Julian Straub, Richard Newcombe, and Steven Lovegrove. DeepSDF: Learning continuous signed distance functions for shape representation. In *Proceedings of the IEEE/CVF Conference on Computer Vision and Pattern Recognition*, pages 165–174, 2019.
- [5] Michal Byra, Charissa Poon, M. F. Rachmadi, Matthias Schlachter, and Henrik Skibbe. Exploring the performance of implicit neural representations for brain image registration. *Scientific Reports*, 13, 2023.
- [6] Kaiyuan Tang and Chaoli Wang. Stsr-inr: Spatiotemporal super-resolution for multivariate time-varying volumetric data via implicit neural representation. *Comput. Graph.*, 119:103874, 2024.
- [7] Hubert Ramsauer, Bernhard Schafkl, Johannes Lehner, Philipp Seidl, Michael Widrich, Lukas Gruber, Markus Holzleitner, Milena Pavlović, Geir Kjetil Ferkingstad Sandve, Victor Greiff, David P. Kreil, Michael Kopp, Günter Klambauer, Johannes Brandstetter, and Sepp Hochreiter. Hopfield networks is all you need. *ArXiv*, abs/2008.02217, 2020.
- [8] Matthew Tancik, Pratul P. Srinivasan, Ben Mildenhall, Sara Fridovich-Keil, Nithin Raghavan, Utkarsh Singhal, Ravi Ramamoorthi, Jonathan T. Barron, and Ren Ng. Fourier features let networks learn high frequency functions in low dimensional domains. *ArXiv*, abs/2006.10739, 2020.
- [9] Matthew Tancik, Pratul P Srinivasan, Ben Mildenhall, Sara Fridovich-Keil, Nithin Raghavan, Utkarsh Singhal, Ravi Ramamoorthi, Jonathan T Barron, and Ren Ng. Fourier features let networks learn high frequency functions in low dimensional domains. In *Advances in Neural Information Processing Systems*, volume 33, pages 7537–7547, 2020.
- [10] Vincent Sitzmann, Julien NP Martel, Alexander W Bergman, David B Lindell, and Gordon Wetzstein. Implicit neural representations with periodic activation functions. *Advances in Neural Information Processing Systems*, 33:7462–7473, 2020.
- [11] Runzhao Yang, Tingxiong Xiao, Yuxiao Cheng, Qi Cao, Jinyuan Qu, Jinli Suo, and Qionghai Dai. Sci: A spectrum concentrated implicit neural compression for biomedical data. *ArXiv*, abs/2209.15180, 2022.
- [12] Runzhao Yang, Tingxiong Xiao, Yuxiao Cheng, Jinli Suo, and Qionghai Dai. Tinc: Tree-structured implicit neural compression. *2023 IEEE/CVF Conference on Computer Vision and Pattern Recognition (CVPR)*, pages 18517–18526, 2022.
- [13] Théophile Blard, Théo Ladune, Pierrick Philippe, Gordon Clare, Xiaoran Jiang, and Olivier Déforges. Overfitted image coding at reduced complexity. *arXiv preprint arXiv:2403.11651*, 2024.
- [14] Hyunjik Kim, Matthias Bauer, Lucas Theis, Jonathan Richard Schwarz, and Emilien Dupont. C3: High-performance and low-complexity neural compression from a single image or video. In *Proceedings of the IEEE/CVF Conference on Computer Vision and Pattern Recognition*, pages 9347–9358, 2024.

- [15] Matheus P Viana, Jianxu Chen, Theo A Knijnenburg, Ritvik Vasani, Calysta Yan, Joy E Arakaki, Matte Bailey, Ben Berry, Antoine Borensztein, Eva M Brown, et al. Integrated intracellular organization and its variations in human ips cells. *Nature*, 613(7943):345–354, 2023.
- [16] Luke Metz, James Harrison, C Daniel Freeman, Amil Merchant, Lucas Beyer, James Bradbury, Naman Agrawal, Ben Poole, Igor Mordatch, Adam Roberts, et al. Velo: Training versatile learned optimizers by scaling up. *arXiv preprint arXiv:2211.09760*, 2022.
- [17] Vivienne Sze, Madhukar Budagavi, and Gary J Sullivan. High efficiency video coding (hevc). 39:40, 2014.
- [18] Charilaos Christopoulos, Athanassios Skodras, and Touradj Ebrahimi. The jpeg2000 still image coding system: an overview. *IEEE transactions on consumer electronics*, 46(4):1103–1127, 2000.
- [19] CL Walsh, P Tafforeau, WL Wagner, DJ Jafree, A Bellier, C Werlein, MP Kühnel, E Boller, S Walker-Samuel, JL Robertus, et al. Imaging intact human organs with local resolution of cellular structures using hierarchical phase-contrast tomography. *Nature methods*, 18(12):1532–1541, 2021.
- [20] Annika Reinke, Minu D Tizabi, Michael Baumgartner, Matthias Eisenmann, Doreen Heckmann-Nötzel, A Emre Kavur, Tim Rädtsch, Carole H Sudre, Laura Acion, Michela Antonelli, et al. Understanding metric-related pitfalls in image analysis validation. *Nature methods*, 21(2):182–194, 2024.
- [21] Jianxu Chen, Matheus P Viana, and Susanne M Rafelski. When seeing is not believing: application-appropriate validation matters for quantitative bioimage analysis. *Nature Methods*, 20(7):968–970, 2023.
- [22] Thomas Wiegand, Gary J Sullivan, Gisle Bjontegaard, and Ajay Luthra. Overview of the h.264/avc video coding standard. *IEEE Transactions on circuits and systems for video technology*, 13(7):560–576, 2003.
- [23] Guo Lu, Wanli Ouyang, Dong Xu, Xiaoyun Zhang, Chunlei Cai, and Zhiyong Gao. Dvc: An end-to-end deep video compression framework. In *Proceedings of the IEEE/CVF conference on computer vision and pattern recognition*, pages 11006–11015, 2019.
- [24] Eirikur Agustsson, David Minnen, Nick Johnston, Johannes Balle, Sung Jin Hwang, and George Toderici. Scale-space flow for end-to-end optimized video compression. In *Proceedings of the IEEE/CVF Conference on Computer Vision and Pattern Recognition*, pages 8503–8512, 2020.
- [25] Hao Chen, Bo He, Hanyu Wang, Yixuan Ren, Ser Nam Lim, and Abhinav Shrivastava. Nerv: Neural representations for videos. *Advances in Neural Information Processing Systems*, 34:21557–21568, 2021.
- [26] Matthew Tancik, Ben Mildenhall, Terrance Wang, Divi Schmidt, Pratul P Srinivasan, Jonathan T Barron, and Ren Ng. Learned initializations for optimizing coordinate-based neural representations. In *Proceedings of the IEEE/CVF Conference on Computer Vision and Pattern Recognition*, pages 2846–2855, 2021.
- [27] Ryan P Abernathy, Tom Augspurger, Anderson Banihirwe, Charles C Blackmon-Luca, Timothy J Crone, Chelle L Gentemann, Joseph J Hamman, Naomi Henderson, Chiara Lepore, Theo A McCaie, et al. Cloud-native repositories for big scientific data. *Computing in Science & Engineering*, 23(2):26–35, 2021.
- [28] Rostam Affendi Hamzah, Muttaqin Md Roslan, Ahmad Fauzan bin Kadmin, Shamsul Fakhar bin Abd Gani, and Khairul Azha A Aziz. Jpg, png and bmp image compression using discrete cosine transform. *TELKOMNIKA (Telecommunication Computing Electronics and Control)*, 19(3):1010–1016, 2021.
- [29] Richard H Wiggins, H Christian Davidson, H Ric Harnsberger, Jason R Lauman, and Patricia A Goede. Image file formats: past, present, and future. *Radiographics*, 21(3):789–798, 2001.

- [30] Ravishankar N Chityala, Kenneth R Hoffmann, Daniel R Bednarek, and Stephen Rudin. Region of interest (roi) computed tomography. *5368:534–541*, 2004.
- [31] Matheus P Viana, Jianxu Chen, Theo A Knijnenburg, Ritvik Vasani, Calysta Yan, Joy E Arakaki, Matte Bailey, Ben Berry, Antoine Borensztein, Eva M Brown, et al. Integrated intracellular organization and its variations in human ips cells. *Nature*, 613(7943):345–354, 2023.
- [32] Hind Taud and Jean-Francois Mas. Multilayer perceptron (mlp). *Geomatic approaches for modeling land change scenarios*, pages 451–455, 2018.
- [33] Vincent Sitzmann, Julien Martel, Alexander Bergman, David Lindell, and Gordon Wetzstein. Implicit neural representations with periodic activation functions. *Advances in neural information processing systems*, 33:7462–7473, 2020.
- [34] Ben Mildenhall, Pratul P Srinivasan, Matthew Tancik, Jonathan T Barron, Ravi Ramamoorthi, and Ren Ng. Nerf: Representing scenes as neural radiance fields for view synthesis. *Communications of the ACM*, 65(1):99–106, 2021.
- [35] Emilien Dupont, Adam Golinski, Milad Alizadeh, Yee Whye Teh, and Arnaud Doucet. Coin: Compression with implicit neural representations. In *Advances in Neural Information Processing Systems*, volume 34, pages 10343–10355, 2021.
- [36] Emilien Dupont, Hrushikesh Loya, Milad Alizadeh, Adam Goliński, Yee Whye Teh, and Arnaud Doucet. Coin++: Neural compression across modalities. *arXiv preprint arXiv:2201.12904*, 2022.
- [37] Jianchen Zhao, Cheng-Ching Tseng, Ming Lu, Ruichuan An, Xiaobao Wei, He Sun, and Shanghang Zhang. Moec: Mixture of experts implicit neural compression. *arXiv preprint arXiv:2312.01361*, 2023.
- [38] Marcin Andrychowicz, Misha Denil, Sergio Gomez, Matthew W Hoffman, David Pfau, Tom Schaul, Brendan Shillingford, and Nando De Freitas. Learning to learn by gradient descent by gradient descent. In *Advances in Neural Information Processing Systems*, pages 3981–3989, 2016.
- [39] Rory M Power and Jan Huisken. A guide to light-sheet fluorescence microscopy for multiscale imaging. *Nature methods*, 14(4):360–373, 2017.
- [40] Juraj Bienik, Miroslav Uhrina, and Peter Kortis. Impact of constant rate factor on objective video quality assessment. *Advances in Electrical and Electronic Engineering*, 15(4):673, 2017.
- [41] Guoqing Xiang, Huizhu Jia, Mingyuan Yang, Yuan Li, and Xiaodong Xie. A novel adaptive quantization method for video coding. *Multimedia Tools and Applications*, 77:14817–14840, 2018.
- [42] Lingling Xu, Haoran Xie, Si-Zhao Joe Qin, Xiaohui Tao, and Fu Lee Wang. Parameter-efficient fine-tuning methods for pretrained language models: A critical review and assessment. *arXiv preprint arXiv:2312.12148*, 2023.
- [43] Luke Metz, James Harrison, C Daniel Freeman, Amil Merchant, Lucas Beyer, James Bradbury, Naman Agrawal, Ben Poole, Igor Mordatch, Adam Roberts, et al. Velo: Training versatile learned optimizers by scaling up. *arXiv preprint arXiv:2211.09760*, 2022.
- [44] GOPAL Patro and Kishore Kumar Sahu. Normalization: A preprocessing stage. *arXiv preprint arXiv:1503.06462*, 2015.

Responses to Reviewers' Comments for Manuscript
NATCOMPUTSCI-24-1168-T

Implicit Neural Image Field for Biological Microscopy Image Compression

Addressed Comments for Publication to
Nature Computational Science

by

Gaole Dai, Cheng-Ching Tseng, Qingpo Wuwu, Rongyu Zhang, Shaokang
Wang, Ming Lu, Tiejun Huang, Yu Zhou, Ali Ata Tuz, Matthias Gunzer, Jianxu
Chen, Shanghang Zhang

Contents

Response to the Meta Review	1
Response to Reviewer 1	6
Response to Reviewer 2	13
Response to Reviewer 3	19

Dear Dr. Ananya Rastogi,

The revised version of our previous revised submission entitled “*Implicit Neural Image Field for Biological Microscopy Image Compression*” with manuscript number **NATCOMPUTSCI-24-1168-T** is enclosed for your review. We would like to express our gratitude to you and the reviewers for providing valuable comments that have greatly enhanced the quality of our manuscript. In this revision, we have diligently addressed all the reviewers’ comments. Reviewer 3 has acknowledged the substantial improvements and robust quantitative foundation of our study, **recommending its publication. All concerns** from Reviewers 1 and 2 in their initial review had been thoroughly addressed in our previous revision. Unfortunately, the newly raised comments by Reviewer 1 and 2 were either false / unfair / unreasonable (we provide detailed justification in our letter) or mostly due to misunderstanding. Therefore, we present a **detailed point-by-point response** to the newly raised comments by Reviewers 1 and 2 (following their order as stated in the decision letter).

Sincerely,

Gaole Dai, Cheng-Ching Tseng, Qingpo Wuwu, Rongyu Zhang, Shaokang Wang, Ming Lu, Tiejun Huang, Yu Zhou, Ali Ata Tuz, Matthias Gunzer, Jianxu Chen, Shanghang Zhang

Note: To enhance the legibility of this response letter, all the editor and reviewers’ comments are typed in boxes. Rephrased or added sentences are typed in colour. The respective parts in the manuscript are highlighted to indicate changes.

Response to the Meta Review

Summary Comment 1

"From the reports, you will see that while they find your work of some potential interest, the referees raise concerns about the advance your findings represent over earlier work and the strength of the novel conclusions that can be drawn at this stage. In particular, they raise concerns about **missing comparisons with SOTA**, the use of **limited microscopy images**, and the **nuclei-touching problem**."

We sincerely appreciate your handling of the review process. Here are our comments regarding the three main points.

1. **Regarding SOTA comparisons:** In our previous revision, we conducted comprehensive benchmarking against 10 established SOTA methods. However, after the revision, **we received further unfair requests from Reviewer 1 and false, unreasonable and out-of-scope requests from Reviewer 2.**
 - Reviewer 1 mentioned the paper by Duan et al. (<https://www.biorxiv.org/content/10.1101/2024.07.17.603794v1>), where a new quantization method proposed in this paper can make AV1 codec achieve good compression performance. But, it is **unfair** to require us to include this method in SOTA comparison, for two reasons: (1) This is still a non-peer-reviewed pre-print. (2) This paper was uploaded to bioRxiv on July 22, 2024. But, our paper was submitted in May 2024, and the pre-print version of our paper was already posted on arXiv on May 29, 2024 (<https://arxiv.org/abs/2405.19012>), which is about 8 weeks before the mentioned pre-print.
 - Reviewer 2 mentioned "quantified results compared with SOTA methods are missing" and "The compression rate is not compared with any SOTA methods.". This is simply **a false judgment**. We conducted thorough comparisons against 10 published SOTA methods in our previous revision, including both published and commercially deployed solutions (JPEG2000 [1], H.264 [2], HEVC [3], DVC [4], NeRF [5], SSF [6], SIREN [7], NeRV [8], SCI [9], TINC [10], Cool-Chic [11], C3 [12]). .
 - Reviewer 2 mentioned "there is only one segmentation task evaluated on Dice". This comment indicates big misunderstanding and the request is **unreasonable and out-of-scope**. Our paper introduces a compression method, not segmentation. Extensive segmentation evaluation is out of the scope of this work. We mentioned the segmentation example, only as to illustrate our concept of application-appropriate validation.

2. **Regarding the diversity of microscopy images:** Reviewer 2 mentioned that "The author uses a minimal amount of microscopy images and of very limited types" and referred that "nnU-Net published in 2021 Nature Method used 23 public datasets". **This claim is simply false and unfair**, for two reasons: (1) The scope of our paper is on microscopy images. Among the 23 datasets used in the nnU-Net paper, there are only **TWO** microscopy examples, CREMI dataset and the cell tracking challenge dataset. (2) Our evaluation dataset selection was strategically designed to encompass the full spectrum of challenges in modern microscopy. The test cases include:
- Dimensional variety: 2D to 5D imaging
 - File sizes: ranging from 100MB to 20GB
 - Multiple modalities: Confocal, Spin-disk Confocal, Dual Oblique Plane, etc.
 - Various imaging conditions: High, Low, and Very Low signal scenarios
 - Diverse biological specimens across species (Mouse, Human, Beetles, etc.)

We have also summarized all these scenarios into a table in [Previous Response Letter Table 4], I hope the reviewer did not miss that. To the best of our knowledge, we are not aware of any method paper systematically evaluated on microscopy datasets of more diverse examples than we did.

3. **Regarding the nuclei-touching concern:** Reviewer 2 mentioned "The innovation in this paper is the contour-aware compression algorithm, as contour recognition is essential in microscopy images. However, it cannot cope with the problems of touching or overlapping nuclei issues or nuclei clusters, that are usually found in microscopy images." There is a **fundamental misunderstanding** of our paper's contributions and methodology. Reviewer 2 **falsely** characterized our work as a "contour-aware compression algorithm" subjected to the "touching-nuclei segmentation problem". Throughout our paper, we never claimed anything related to "contour-aware compression". Reviewer 2's statement "The compression performance is subject to the capability of nuclei segmentation in the proposed method." is simply **unfair** and **unreasonable**. For example, when two nuclei in the original microscopy are touching and there is barely any clear separation boundary, this is the problem of the microscopy image acquisition, e.g., limited resolution, which could be resolved using a higher-mag lens (e.g., switching from 20x to 100x). There is nothing we could do in the compression step. It is **unfair** to blame the compression algorithm for not able to separate the touching nuclei.

We have compiled a comprehensive checklist (Fig. 1, see below) of all concerns raised across both review rounds. While Reviewer 3 has expressed complete satisfaction with our revisions, we remain committed to addressing every remaining point in detail in the following sections. However, we feel obligated to note that some requests fall outside the reasonable scope of our work or are technically unfair / unreasonable. In such cases, we provide clear explanations for these limitations and suggest practical alternatives where possible.

- [1] C. Christopoulos, A. Skodras, and T. Ebrahimi, "The jpeg2000 still image coding system: An overview," *IEEE transactions on consumer electronics*, vol. 46, no. 4, pp. 1103–1127, 2000.
- [2] T. Wiegand, G. J. Sullivan, G. Bjontegaard, and A. Luthra, "Overview of the h. 264/avc video coding standard," *IEEE Transactions on circuits and systems for video technology*, vol. 13, no. 7, pp. 560–576, 2003.
- [3] V. Sze, M. Budagavi, and G. J. Sullivan, "High efficiency video coding (hevc)," vol. 39, p. 40, 2014.
- [4] G. Lu, W. Ouyang, D. Xu, X. Zhang, C. Cai, and Z. Gao, "Dvc: An end-to-end deep video compression framework," in *Proceedings of the IEEE/CVF conference on computer vision and pattern recognition*, 2019, pp. 11 006–11 015.
- [5] B. Mildenhall, P. P. Srinivasan, M. Tancik, J. T. Barron, R. Ramamoorthi, and R. Ng, "Nerf: Representing scenes as neural radiance fields for view synthesis," in *European Conference on Computer Vision*, Springer, 2020, pp. 405–421.
- [6] E. Agustsson, D. Minnen, N. Johnston, J. Balle, S. J. Hwang, and G. Toderici, "Scale-space flow for end-to-end optimized video compression," in *Proceedings of the IEEE/CVF Conference on Computer Vision and Pattern Recognition*, 2020, pp. 8503–8512.
- [7] V. Sitzmann, J. N. Martel, A. W. Bergman, D. B. Lindell, and G. Wetzstein, "Implicit neural representations with periodic activation functions," *Advances in Neural Information Processing Systems*, vol. 33, pp. 7462–7473, 2020.
- [8] H. Chen, B. He, H. Wang, Y. Ren, S. N. Lim, and A. Shrivastava, "Nerv: Neural representations for videos," *Advances in Neural Information Processing Systems*, vol. 34, pp. 21 557–21 568, 2021.
- [9] R. Yang, T. Xiao, Y. Cheng, *et al.*, "Sci: A spectrum concentrated implicit neural compression for biomedical data," *ArXiv*, vol. abs/2209.15180, 2022.
- [10] R. Yang, T. Xiao, Y. Cheng, J. Suo, and Q. Dai, "Tinc: Tree-structured implicit neural compression," *2023 IEEE/CVF Conference on Computer Vision and Pattern Recognition (CVPR)*, pp. 18 517–18 526, 2022.

- [11] T. Blard, T. Ladune, P. Philippe, G. Clare, X. Jiang, and O. Déforbes, “Overfitted image coding at reduced complexity,” *arXiv preprint arXiv:2403.11651*, 2024.
- [12] H. Kim, M. Bauer, L. Theis, J. R. Schwarz, and E. Dupont, “C3: High-performance and low-complexity neural compression from a single image or video,” in *Proceedings of the IEEE/CVF Conference on Computer Vision and Pattern Recognition*, 2024, pp. 9347–9358.

Figure 1: We summarized the comments from 1st and 2nd round review, finding majorities of concerns are solved by our previous revision

Response to Reviewer 1

Comment 1.1

"The poor performance of HEVC when targeted to a 256x compression ratio is caused by non-optimal CODEC settings."

Response 1.1:

We respectfully disagree with this assessment. Our experimental methodology involved extensive hyperparameter optimization, thoroughly documented in [Manuscript lines 146-161] and [Previous Response Letter Section 2.3]. Specifically:

1. We conducted comprehensive grid searches across multiple parameter spaces for each CODEC, ensuring optimal performance configurations.
2. The inherent variability in CODEC compression ratios highlights a key advantage of our INIF approach - the ability to achieve precise target compression ratios in a single pass, eliminating the need for iterative parameter tuning.
3. Our experiments systematically evaluated both controlled compression scenarios and real-world scenarios where CODECs failed to meet target ratios while quantifying the often-overlooked computational overhead of hyperparameter optimization [Previous Response Letter Section 1.2.2].

Manuscript: Line 148-161

Furthermore, even with meticulous adjustment of hyper-parameters such as bitrate [1], constant rate factor (CRF) [2], and adaptive quantization (AQ) mode [3], it is unable to control the compression rate throughout the HEVC experiment. In contrast, our proposed INIF method consistently maintains clear visibility of cellular contours and retains considerable textural details under controllable high compression ratios of 128 and 256-fold.

- [1] R. M. Power and J. Huisken, "A guide to light-sheet fluorescence microscopy for multiscale imaging," *Nature methods*, vol. 14, no. 4, pp. 360–373, 2017.
- [2] J. Bienik, M. Uhrina, and P. Kortis, "Impact of constant rate factor on objective video quality assessment," *Advances in Electrical and Electronic Engineering*, vol. 15, no. 4, p. 673, 2017.

- [3] G. Xiang, H. Jia, M. Yang, Y. Li, and X. Xie, "A novel adaptive quantization method for video coding," *Multimedia Tools and Applications*, vol. 77, pp. 14 817–14 840, 2018.

Previous Response: Section 2.3

We appreciate the reviewer's concern regarding the fairness of comparisons, particularly with HEVC. We have addressed this issue in several ways:

- (a) As mentioned in our manuscript lines [148-161], one of the major drawbacks of traditional CODECs is the difficulty in precisely controlling compression ratios. To be specific, the "meticulous adjustment" we conducted is a hyperparameter sweep for our CODEC competitors, including Bitrate, Constant Rate Factor (CRF), and Adaptive Quantization (AQ) mode, with a search depth of 10. Bitrate directly controls the output file size, CRF balances compression efficiency and video quality, and AQ-mode adjusts quantization strength based on frame complexity. The combination of these parameters allows for fine-grained control over the compression process, theoretically enabling ratio management to some extent. We performed this grid search process for each sample to find a reasonable parameter combination. However, despite this extensive parameter tuning, there are still scenarios where precise control of compression ratios remains challenging for traditional CODECs. We deliberately included such cases in our study to highlight this limitation. For instance, in Figures 2 and 6 of our manuscript, we showcase examples where HEVC struggles to achieve the target compression ratio, even after our thorough hyperparameter optimization. These examples underscore the inherent variability in CODEC performance across different types of microscopy data.
- (b) In most cases, we have successfully matched the compression ratios between INIF and other methods. This is evident in Figures 3, 4, 5, 6, and 7 of the main text, where comparisons are made at equivalent compression ratios.
- (c) In Figure 4, we demonstrate that INIF achieves better results than HEVC on high-dimensional data compression tasks while using 4-fold less bitrate.
- (d) In cases where exact ratio matching was not possible (e.g., Figures 2 and 6), we emphasize that HEVC can face severe failures with certain types of data, such as noisy images or images with drastic changes

between frames/depth. In contrast, INIF, rooted in implicit neural representations, offers a more robust compression approach for diverse data types.

Previous Response: Section 1.2.2

Another cost which a learned optimizer can significantly reduce is the cost of searching hyperparameters. This cost is important in compressing tasks since we unavoidably need a set of hyperparameters for each sample no matter whether using CODEC or INR to get a close to optimal outcome. However, this cost is often ignored. For CODEC, a range of hyperparameters controls the compression ratio and affects the final quality. For example, we have searched for the Bitrate, Constant Rate Factor (CRF), and Adaptive Quantization (AQ) mode when using CODEC. For INR-based methods, the basic hyperparameter for all the baselines is the learning rate (LR) and additional hyperparameters for each method specifically e.g. GOP mode for Cool-chic, partition number for TINC etc. We have demonstrated improper LR would severely affect the final compression quality. This would be a bigger issue when compressing large data. Where, the cost of searching a set of reasonable hyperparameters would be fairly high because this requires multiple search trails. INIF, in contrast, uses a learned optimizer to search for the optimal optimization direction and step size automatically. Which does not need to set a hard hyperparameter such as the LR before training. All the results shown are based on a single trial, this could be considered as a very convenient feature for further propagating INR-based compressors as a usable tool to the community.

Comment 1.2

"Duan et al.(<https://www.biorxiv.org/content/10.1101/2024.07.17.603794v1>) showed that AV1 CODEC can compress similar quality nucleus images by up to 12,000x with visually much more faithful results than the 256x compressed results presented in this manuscript."

Response 1.2:

We appreciate the reviewer bringing this work to our attention. However, several critical factors make this comparison scientifically inappropriate:

1. The referenced work is a **non-peer-reviewed preprint** posted in July 2024, after our submission and initial review process.
2. The study uses fundamentally **different datasets**, making direct visual

quality comparisons methodologically unsound.

3. The **absence of open-source code** and comprehensive methodology details precludes rigorous comparative analysis.
4. Most importantly, our INIF framework is designed to be **complementary** to CODEC improvements, not competitive. The advances in CODEC optimization could potentially be integrated into our framework, further enhancing overall performance.

Comment 1.3

"The statement that using a model that allows pixel-level image synthesis will lead to faster I/O performance when handling large-scale images is misleading. This is because when the image is large, the model size will be exponentially increased to compensate for the image dimensions. As it always requires the whole model to reconstruct even a single pixel, the time and memory required for loading the whole model can be slow or even impossible."

Response 1.3:

We appreciate this concern but must respectfully disagree with the underlying assumptions. As detailed in [Manuscript Appendix D] and [Previous Response Letter Section 1.3], our analysis reveals several key insights:

1. **Model Scaling:** The relationship between model size and data volume is not exponential but rather follows a more nuanced scaling pattern, modulated by the target compression ratio and architectural design choices.
2. **Empirical Evidence:** We conducted a benchmark test and the performance analysis (see Fig 7) demonstrated that INIF achieves superior decoding efficiency, particularly for high-dimensional large data, due to:
 - Optimized network architecture with minimal partitioning
 - Efficient parallel processing capabilities
 - Reduced computational overhead compared to traditional approaches
3. **Technical Advantages:**
 - Unlike sequential decoders (common in CODECs and autoregressive models), INIF enables random-access parallel decoding for arbitrary regions of interest

- The framework supports flexible trade-offs between model complexity and reconstruction quality by minimizing the cost of hyperparameter search when a specific compression ratio is assigned.

These characteristics make INIF particularly well-suited for real-world applications requiring efficient handling of large-scale image data while maintaining competitive compression ratios and reconstruction quality.

Manuscript: Appendix D

Commonly used file formats (e.g., JPG and PNG) can only be used for regular 2D RGB images. Specialized file formats such as Ome-TIFF or Zarr have been developed for storing and decoding high-dimensional microscopy images [4]–[6]. Compared to Ome-Tiff, Zarr [4] offers more flexibility with additional features like multi-resolution storage, block-wise decoding, and parallel decoding using GPU. The evolution of these file formats indicates that the ultimate objective is to efficiently decode and visualize specific regions of interest (ROI) [7] in any microscopy data as needed since full decoding can be time-consuming and memory-intensive. In this context, INIF provides an excellent solution by enabling pixel-wise decoding. Taking a 3D (XYZ) confocal microscopy image of DNA [8] as an example, assume our goal is to visualize slice 32. Decoding TIFF requires complete decoding of the entire dataset with dimensions X:924, Y:624, and Z:65 – including extracting all pixel values followed by selecting only those belonging to slice 32 for visualization purposes. On the other hand, leveraging INIF’s pixel-wise decoding function allows us to simply provide the coordinates of slice 32 to the INR network for efficient decoding. Furthermore, by assigning a subset of coordinates, we can also support multi-resolution functionality – particularly useful when fast preview-based decodings are required without needing additional storage space for different resolution versions of the same data which is used in Zarr format. Additionally, even in special scenarios involving irregularly shaped ROIs represented by binary masks, we can still handle them through appropriate location-based decoding (Fig 6).

- [4] R. P. Abernathy, T. Augspurger, A. Banihirwe, *et al.*, “Cloud-native repositories for big scientific data,” *Computing in Science & Engineering*, vol. 23, no. 2, pp. 26–35, 2021.
- [5] R. A. Hamzah, M. M. Roslan, A. F. bin Kadimin, S. F. bin Abd Gani, and K. A. A. Aziz, “Jpg, png and bmp image compression using discrete cosine transform,” *TELKOMNIKA (Telecommunication Computing Electronics and Control)*, vol. 19, no. 3, pp. 1010–1016, 2021.

- [6] R. H. Wiggins, H. C. Davidson, H. R. Harnsberger, J. R. Lauman, and P. A. Goede, "Image file formats: Past, present, and future," *Radiographics*, vol. 21, no. 3, pp. 789–798, 2001.
- [7] R. N. Chityala, K. R. Hoffmann, D. R. Bednarek, and S. Rudin, "Region of interest (roi) computed tomography," vol. 5368, pp. 534–541, 2004.
- [8] M. P. Viana, J. Chen, T. A. Knijnenburg, *et al.*, "Integrated intracellular organization and its variations in human ips cells," *Nature*, vol. 613, no. 7943, pp. 345–354, 2023.

Figure 6: Decoding INIF file in a pixel-wise fashion. Unlike traditional file formats like TIFF that require complete decoding of the entire dataset, INIF enables pixel-wise decoding by providing the coordinates of the desired region of interest (ROI) to the INR network. INIF also supports multi-resolution decoding functionality and location-specific decoding even for irregularly shaped ROIs represented by binary masks.

Figure 7: Compression performance comparison of various methods across different data dimensions. The graphs illustrate the relative costs of different methods over HEVC. INIF demonstrates consistent performance across dimensions, particularly excelling in high-dimensional scenarios.

Response to Reviewer 2

Comment 2.1

"The images from Figure 2 to Figure 7 are all the demonstrations of particular cases. The profile and visualization of the whole dataset are missing. Moreover, quantified results compared with SOTA methods are missing. For example, in Section 1.2.3, the manuscript says 'using simple metrics such as PSNR or SSIM often contains bias and can be misleading when dealing with biomedical data', but there is only one segmentation task evaluated on Dice."

Response 2.1:

The statement about limited case demonstrations and missing comparisons is inaccurate. We have provided comprehensive dataset-level analyses and extensive comparisons with SOTA methods throughout our manuscript and response letter:

1. Dataset-level analyses are presented in: [**Manuscript: Figures 2, 4, and 6**] and [**Previous Response Letter: Tables 2, 4, and 5**]. Specifically, The dataset used in Figure 2 named human induced pluripotent stem cell (hiPSC) [1] contains 25 types of cell lines, we randomly selected 1 sample from each cell line to construct our subset for testing. Figure 4 directly uses the dataset from the Nature Methods paper CARE [2], the link to the dataset is <https://publications.mpi-cbg.de/publications-sites/7207/>. Figure 5 used the data <https://zenodo.org/records/7544194> from the Biophysics Journal paper by Smith et al. [3], originally for 5D organoid tracking.
2. We conducted thorough comparisons against 10 published SOTA methods (JPEG2000, H.264, HEVC, DVC, NeRF, SSF, SIREN, NeRV, SCI, TINC, CoolChic, C3)

The assertion that "only one segmentation task evaluated on Dice" misrepresents our work. In addition to this evaluation metric for segmentation tasks mentioned by the reviewer, we also have a full section demonstrating limitations of PSNR/SSIM metrics in another context - Low laser microscopy compression analysis - which can be found in [**Manuscript Section 2.2**] and [**Previous Response Letter Figure 7**] (see Fig 8). This case highlights why conventional metrics like PSNR/SSIM can be misleading for biological image compression; thus necessitating application-appropriate evaluation metrics. By adopting a comprehensive evaluation approach across multiple scenarios and metrics, we provide a more robust assessment of compression performance than relying solely on traditional metrics.

Figure 8: Naive metrics are not always robust and even misleading when dealing with biomedical images.

- [1] M. P. Viana, J. Chen, T. A. Knijnenburg, *et al.*, "Integrated intracellular organization and its variations in human ips cells," *Nature*, vol. 613, no. 7943, pp. 345–354, 2023.
- [2] M. Weigert, U. Schmidt, T. Boothe, *et al.*, "Content-aware image restoration: Pushing the limits of fluorescence microscopy," *Nature methods*, vol. 15, no. 12, pp. 1090–1097, 2018.
- [3] M. B. Smith, H. Sparks, J. Almagro, *et al.*, "Active mesh and neural network pipeline for cell aggregate segmentation," *bioRxiv*, 2023. DOI: 10.1101/2023.02.17.528925. eprint: <https://www.biorxiv.org/content/early/2023/02/21/2023.02.17.528925.full.pdf>.

Comment 2.2

"The innovation in this paper is the contour-aware compression algorithm, as contour recognition is essential in microscopy images. However, it cannot cope with the problems of touching or overlapping nuclei issues or nuclei clusters, that are usually found in microscopy images."

Figure 9: The raw data quality would effect the compression result.

Comment 2.3

"The compression performance is subject to the capability of nuclei segmentation in the proposed method. Moreover, inner areas are also of high importance."

Response 2.3:

There appears to be a fundamental misunderstanding of our paper's contributions and methodology. The incorrect characterizes of our work as a "contour-aware compression algorithm" subjected to "nuclei segmentation", are never claimed or addressed in our manuscript. Our innovations present a complete INR-based compression workflow for biological images, where application-appropriate guidance serves as one flexible component rather than being limited to contour recognition. Our method neither performs nor depends on nuclei segmentation, and in its default operation, treats all pixels equally without bias toward inner or outer regions. Through the application-appropriate guidance mechanism, users can optionally specify ROI for focused compression, providing flexibility for different experimental needs.

The example of overlapping nuclei in relatively low-resolution raw images was presented in our Limitations Section [**Manuscript Appendix F**] to illustrate the distinct characteristics of artifacts between INR compression and traditional CODECs. Unlike block artifacts typically found in CODECs, INR compression produces oversmooth artifacts. This differentiation is crucial for comprehending compression choices at various resolutions, and we explicitly acknowledge that lossless compression should be utilized for critical applications where any form of artifact is deemed unacceptable. Here we provide a example when compressing higher resolution nuclei image in Figure 9, we can see if the raw image has clear contour, the output after INIF's compression could also be satisfying.

Comment 2.4

"This is more applicable to journals like IEEE Transactions of Medical Imaging. A general model should take in many more datasets. For example, "nnU-Net" published in 2021 Nature Method used "23 public datasets used in international biomedical segmentation competitions." The biomedical dataset boosted in the past 3 years."

Response 2.4:

The suggestion to compare our dataset coverage with nnU-Net reflects a misunderstanding of both our work's scope and the fundamental differences between image compression and segmentation tasks. Our work specifically targets biological image compression, rather than a general model. We have deliberately chosen datasets that represent the diverse challenges in modern microscopy. Our test cases span dimensions from 2D to 5D, with individual file sizes ranging from 100MB to 20GB, encompassing various imaging modalities, imaging conditions, and biological specimens across different species. The comparison to nnU-Net's 23 datasets is inappropriate not only because it addresses a different task (segmentation versus compression), but also because of fundamental differences in data characteristics. While nnU-Net's datasets are limited to 2D and 3D images with only two datasets being microscopy, our work tackles the more challenging aspects of high-dimensional, large-scale biological imaging data that better represents the current challenges in the compression field.

Comment 2.5

"The compression ratio (128X, 256X) is compared with the original raw images. The compression rate is not compared with any SOTA methods. Comprehensive rate-distortion performance metrics and comparisons are required to validate the effectiveness. Specifically, the authors presented 'This dataset serves as the benchmark test for both SCI and TINC and they are representative state-of-the-art INR-based compressors for specialized biomedical data'. Yet the comparison is missing in Table 4. The performance gains from the INIF adapter appear heavily contingent on the use of specific CODECs like HEVC. Please compare the performance with alternative CODECs, such as JPEG2000."

Response 2.5:

The concern about limited comparisons appears to overlook substantial evidence presented in our manuscript and previous response letter. We have consistently provided comprehensive comparisons against both raw images and

Figure 10: Compression Rate-distortion curve of INIF, SCI, and TINC with HipCT Dataset.

multiple baselines throughout our experiments. Specifically, in [Previous Response Letter Figure 2, Table 5, 6], we presented detailed comparisons against more than ten published SOTA and commonly used compression methods. The apparent absence of SCI and TINC results simply reflects that Table 4 was specifically designed to provide quantitative metrics for the qualitative examples shown in the manuscript - we already showed comprehensive quantitative results in [Previous Response Letter Table 5], and we are certainly willing to supplement this table with rate-distortion performance metrics (see Fig. 10). Furthermore, the suggestion that the INIF adapter’s performance is heavily dependent on HEVC is incorrect. As demonstrated in [Previous Response Letter Figure 3], our method achieves comparable performance even without CODEC priors, and we can readily incorporate other CODECs as priors if needed. This adaptability highlights the flexibility of our approach rather than a dependency on any specific CODEC, where we provide additional experimental results with alternative JPEG2000 CODECs to further demonstrate this point (see Fig 11).

Figure 11: Convergent speed and Overall performance comparison with different residual.

Response to Reviewer 3

Comment 3.1

The authors have submitted a revised version of their manuscript, and I greatly appreciate the effort they have invested in addressing all the referee comments and suggestions. I have carefully reviewed the resubmission. The authors have responded thoroughly and thoughtfully to the feedback, including my own, and have made significant improvements. These revisions have notably enhanced both the clarity and rigour of the manuscript. In particular, the quantitative comparison now provides a much stronger foundation for the paper's conclusions.

Response 3.1:

We sincerely thank the reviewer for their thorough evaluation and constructive feedback throughout the review process. Your insightful comments and suggestions have been instrumental in improving the quality and clarity of our manuscript. The rigorous review process has helped us strengthen our quantitative analysis, refine our methodology, and present our findings more effectively. We particularly appreciate your attention to detail and thoughtful consideration of our revisions. Your guidance has significantly contributed to elevating the scientific rigour of our work and enhancing its potential impact in the field.

Responses to Reviewers' Comments for Manuscript
NATCOMPUTSCI-24-1168-T

Implicit Neural Image Field for Biological Microscopy Image Compression

Addressed Comments for Publication to
Nature Computational Science

by

Gaole Dai, Cheng-Ching Tseng, Qingpo Wuwu, Rongyu Zhang, Shaokang
Wang, Ming Lu, Tiejun Huang, Yu Zhou, Ali Ata Tuz, Matthias Gunzer, Jianxu
Chen, Shanghang Zhang

Contents

Response to the Meta Review	1
Response to Reviewer 2	2

Dear Dr. Ananya Rastogi,

The revised version of our prior submission, titled “*Implicit Neural Image Field for Biological Microscopy Image Compression*”, with manuscript number **NATCOMPUTSCI-24-1168-T**, is enclosed for your review. We sincerely appreciate the constructive feedback provided by both you and the reviewers, which has substantially enhanced the quality of our work. In this revision, we have meticulously addressed the remaining concern. Specifically, we have conducted a thorough review in response to Review 2’s suggestion regarding the inclusion of domain-specific evaluation metrics, such as the DICE score for segmentation. Metric such as DICE, LPIPS, and energy usage was already reported in **Table 1** of the **previous Revised Manuscript** and detailed in the Response Letter. Nevertheless, to further clarify and substantiate our findings, we have incorporated and appended additional results and comprehensive case studies to the manuscript, which we believe fully address the reviewer’s comments.

Sincerely,

Gaole Dai, Cheng-Ching Tseng, Qingpo Wuwu, Rongyu Zhang, Shaokang Wang, Ming Lu, Tiejun Huang, Yu Zhou, Ali Ata Tuz, Matthias Gunzer, Jianxu Chen, Shanghang Zhang

Note: To enhance the legibility of this response letter, all the editor and reviewers’ comments are typed in boxes. Rephrased or added sentences are typed in color. The respective parts in the manuscript are highlighted to indicate changes.

Response to the Meta Review

Summary Comment 1

"To ensure real-world applicability, please validate performance on downstream tasks (e.g., DICE scores of segmentation on compressed data) or discuss this as a limitation."

We sincerely appreciate your diligent handling of the review process and are delighted to address the reviewers' concerns in this revised submission.

We fully recognize the significance of demonstrating the real-world applicability of our proposed method. To this end, **we have already validated the performance of INIF on the downstream segmentation task, reporting DICE scores compared to baseline methods in Table 1 of our previously revised manuscript.** In this response, we have gone a step further to provide additional case studies and results to comprehensively highlight the advantages of INIF. These new analyses offer more profound insights into our method's practical utility across diverse scenarios, further underscoring its superiority.

Response to Reviewer 2

Comment 1.1

To ensure real-world applicability, it is important to validate performance on downstream tasks (e.g., DICE scores of segmentation on compressed data). Authors are strongly encouraged to either (1) provide sufficient quantitative downstream task analyses (even using simple baselines) or (2) acknowledge this limitation and discuss its potential impact on biological applications in the revised paper.

Response 1.1:

We sincerely thank the reviewer for their insightful comments, which emphasize the importance of validating the real-world applicability of our method through downstream task analyses. We have carefully addressed these concerns and made substantial improvements in the revised manuscript, as below:

In our previously revised manuscript, we reported DICE scores in Table 1, demonstrating the performance of INIF when segmentation guidance was incorporated. These results already highlight the effectiveness of INIF in preserving segmentation accuracy on compressed data. However, we fully recognize the importance of providing additional analyses to further validate the practical utility of our method, particularly in scenarios where maintaining semantic fidelity is critical, such as microscopy imaging. To this end, we have included new case studies in the response (see Figure 1 and Figure 2), further illustrating how INIF achieve superior performance in real-world applications.

1. For example, in *prompt-based segmentation* leveraging Segment Anything Model (SAM) [1], the perceptual quality of compressed images plays a pivotal role in enabling users to achieve specific segmentation objectives through prompts. This dual requirement—preserving model perceptual fidelity and human perceptual similarity—represents a unique challenge in real-world scenarios. *Dense grid point prompting* is a well-established technique for performing panoramic segmentation using the SAM model. Each grid intersection point serves as a prompt for SAM, generating a unique output mask. Subsequently, de-duplication is performed among the masks to produce the final mask. Our results under such settings (Figure 1) illustrate that INIF’s compression technique effectively retains SAM’s perceptual similarity, leading to accurate and reliable segmentation outcomes, even when compression is applied. This is particularly critical for microscopy images, where even subtle deviations in model perceptual similarity can result in significant segmentation errors, which are also hard to identify by metrics alone.

2. Furthermore, in *interactive sparse box prompting* scenarios (Figure 2), which is also the most common usage in SAM. Users can employ three types of prompts (point, box, and mask) to control the masking output according to specific demands. We tested the use of box prompts. INIF shows its capability to avoid severe block artifacts or the oversmoothed blending of adjacent structures during compression, issues commonly observed in competing methods. As a result, INIF ensures segmentation consistency across different scales (i.e, foreground-background, extracting a single cell), allowing users to identify the target prompting area and achieve desired results regardless of the granularity of their segmentation requirement. This is important in applications where maintaining the structural integrity of individual cells is crucial for downstream biological analysis.

We believe these additional case studies and the improvements made in the revised manuscript comprehensively address the reviewer's concerns. By showcasing the advantages of INIF in dense and sparse prompted segmentation scenarios with SAM, we highlight its ability to handle diverse and challenging real-world tasks in microscopy imaging and beyond. Thank you again for your constructive feedback, which has helped us significantly strengthen our work.

- [1] A. Kirillov, E. Mintun, N. Ravi, *et al.*, "Segment anything," in *Proceedings of the IEEE/CVF international conference on computer vision*, 2023, pp. 4015–4026.

Figure 1: Dense Prompted Result with Segment Anything Model, To facilitate comparison, we have highlighted the common failure cases in other compression outcomes. Columns 1 and 3 (from left) illustrate the compressed images produced by various compressors when compared to the raw image. Columns 2 and 4 (from left) present the corresponding segmentation results after dense grid prompting. Implicit methods, such as SIREN and INIF, demonstrate superior performance; however, INIF exhibits better handling of corner cases when the edge contrast of nuclei is low.

Figure 2: Sparse Prompted Result with Segment Anything Model. To demonstrate various demand scenarios, we focused on the segmentation of a single cell as well as the separation of foreground and background. The first column (from left) presents the compression result in comparison to the raw image. The second and fourth columns (from left) illustrate the box prompts with varying scales provided to the SAM model. The third and fifth columns (from left) compare the segmentation results. The yellow mask is the pseudo ground truth (GT) mask obtained from the raw image via SAM. The original raw image is then cut out by the segmentation results from either the original image or the compressed images and overlaid with the yellow pseudo GT mask, in order to visualize the consistency between the segmentation results and the pseudo GT over the original image. Our INIF method exhibited the highest consistency across both scenarios, achieving superior DICE scores and visualization fidelity.